# A novel *Drosophila* injury model reveals severed axons are cleared through a Draper/MMP-1 signaling cascade

Maria D Purice[1], Arpita Ray[1], Eva Jolanda Münzel[1], Bernard J Pope[2], Daniel J Park[2], Sean D Speese[1]*, Mary A Logan[1]*

[1]Jungers Center for Neurosciences Research, Department of Neurology, Oregon Health and Science University, Portland, United States; [2]Melbourne Informatics, The University of Melbourne, Melbourne, Australia

**Abstract** Neural injury triggers swift responses from glia, including glial migration and phagocytic clearance of damaged neurons. The transcriptional programs governing these complex innate glial immune responses are still unclear. Here, we describe a novel injury assay in adult *Drosophila* that elicits widespread glial responses in the ventral nerve cord (VNC). We profiled injury-induced changes in VNC gene expression by RNA sequencing (RNA-seq) and found that responsive genes fall into diverse signaling classes. One factor, matrix metalloproteinase-1 (MMP-1), is induced in *Drosophila* ensheathing glia responding to severed axons. Interestingly, glial induction of MMP-1 requires the highly conserved engulfment receptor Draper, as well as AP-1 and STAT92E. In MMP-1 depleted flies, glia do not properly infiltrate neuropil regions after axotomy and, as a consequence, fail to clear degenerating axonal debris. This work identifies Draper-dependent activation of MMP-1 as a novel cascade required for proper glial clearance of severed axons.

*For correspondence: speese@ohsu.edu (SDS); loganm@ohsu.edu (MAL)

**Competing interests:** The authors declare that no competing interests exist.

## Introduction

Glial cells exhibit swift and dramatic responses to any form of neural trauma. These reactions provide neuroprotection and minimize further damage to the central nervous system (CNS. Changes in glial gene expression is one highly conserved feature of glia responding to a range of insults in the CNS (*Allen and Barres, 2009*; *Chung et al., 2013*; *Doherty et al., 2009*; *Logan and Freeman, 2007*; *Logan et al., 2012*; *Ziegenfuss et al., 2008*). Specifically, the janus kinase/signal transducer and activator of transcription (JAK/STAT) cascade is one central player in initiating reactive astrocytic and microglial responses to neural damage (*Ben Haim et al., 2015*; *Herrmann et al., 2008*; *Kim et al., 2002*; *Park et al., 2016*). Glial activation of the c-Jun N-terminal kinase (JNK) pathway and the downstream transcriptional heterodimer AP-1, which consists of c-Fos and c-Jun, has also been reported in various injury and disease models (*Anderson et al., 1994*; *Pennypacker et al., 1994*; *Yu et al., 1995*). However, it remains unclear what pathways are required to initiate these transcriptional programs and how they drive complex glial responses to neural injury.

Reactive glia often display prominent changes in cell shape, size, or motility in response to neuronal damage. Glial cells can migrate substantial distances to reach trauma sites (*Roth et al., 2014*). In other instances, glial somas remain in fixed locations while the cells extend elongated processes to rapidly enter regions that house damaged neurons (*Davalos et al., 2005*; *Dissing-Olesen et al., 2014*). These striking morphogenic responses ensure that glial cells gain access to sites of damage to release protective (and sometimes detrimental) factors and clear apopototic cells and degenerating projections through phagocytic engulfment (*Hines et al., 2009*; *Polazzi and Monti, 2010*), but

we still have an incomplete understanding of how these dynamic responses are elicited and carried out in reactive glia.

Glial responses to injury and disease share common features across species. In *Drosophila*, acute axotomy triggers reactions from local glia that are strikingly similar to those observed in mammals. For example, severing adult *Drosophila* olfactory nerves that project into the antennal lobes of the central brain initiates a classic Wallerian degeneration (WD) program in olfactory receptor neuron (ORN) axons (*Hoopfer et al., 2006*; *MacDonald et al., 2006*). Over the course of several days, local glia extend membrane projections into the antennal lobe neuropil to phagocytose degenerating axonal and synaptic debris (*Logan and Freeman, 2007*; *Logan et al., 2012*; *MacDonald et al., 2006*; *Ziegenfuss et al., 2012*). Common molecular pathways are also activated in fly and mammalian glia responding to injury (*Awasaki et al., 2006*; *Doherty et al., 2014*; *Lu et al., 2014*; *Macdonald et al., 2013*; *Ziegenfuss et al., 2008*). As in mammals, the transcription factors AP-1 and STAT92E are acutely induced in *Drosophila* ensheathing glia in response to axotomy (*Doherty et al., 2014*; *Macdonald et al., 2013*). In addition, common molecules are required for glial clearance of engulfment targets across species, including the conserved Draper/MEGF10 receptor, downstream Syk tyrosine kinases, and Rac1-mediated cytoskeletal remodeling (*Awasaki et al., 2006*; *Doherty et al., 2009*; *Logan and Freeman, 2007*; *Logan et al., 2012*; *MacDonald et al., 2006*; *Scheib et al., 2012*). Thus, the fly represents a powerful genetic model system to explore new molecular cascades and probe the basic mechanisms driving innate glial immune reactions.

In adult flies, axotomy triggers activation of the Draper receptor on local ensheathing glia, which, in turn, stimulates a positive auto-regulatory transcriptional loop. Specifically, STAT92E becomes activated and targets the *draper* locus to upregulate *draper-I* transcript levels (*Doherty et al., 2014*). This response ensures that adequate levels of Draper are present at the cell surface of phagocytic reactive glia. Aside from STAT92E targeting of *draper*, virtually nothing is known about the transcriptional programs invoked in adult *Drosophila* glia responding to neural injury. Here, we describe a new non-lethal nerve injury model which elicits widespread glial responses to neurodegeneration in the ventral nerve cord (VNC). Using this assay, we carried out an RNA-sequencing (RNA-Seq) analysis of VNC tissue to identify axotomy-induced changes in gene expression. We also performed a detailed in vivo functional analysis of the secreted protease matrix metalloproteinase-1 (MMP-1), a novel gene we found to be acutely upregulated in ensheathing glia responding to nerve injury. Interestingly, we show that MMP-1 induction in ensheathing glia requires Draper/STAT92E/AP-1 signaling. MMP-1 is required for proper glial recruitment to severed axons and, subsequently, timely clearance of degenerating axonal debris. Together, this work demonstrates an essential role for MMP-1 in glial recruitment and phagocytic responses post-axotomy and reveals a novel connection between Draper/STAT92E/AP-1 and requisite MMP-1 expression during glial responses to nerve injury.

## Results

### Draper is robustly upregulated in VNC ensheathing glia after axotomy

In the adult *Drosophila* olfactory system, local ensheathing glial cells upregulate the pro-engulfment isoform *draper-I* in an AP-1/STAT92E-dependent manner in response to axotomy, suggesting that neurodegeneration triggers transcriptional programs in fly glia. However, the glial cells that surround antennal lobe neuropil (*Figure 1a,b* antennal lobe, orange circles) represent a small percentage of total brain glia, which presents substantive challenges in assessing changes in antennal glial gene expression post-injury. We sought to develop an in vivo injury model that would elicit a broader glial response in order to identify differentially expressed genes using a large-scale transcriptional profiling screen. Recent studies revealed that peripheral sensory neurons in the wing undergo a classic Wallerian degeneration program and are cleared by glia in a Draper-dependent manner after axotomy (*Fang et al., 2012*; *Fullard and Baker, 2015*), suggesting that glial cells throughout the adult fly nervous system use common pathways to sense and respond to degenerating axons. Thus, we turned to the ventral nerve cord (VNC) of the fly with the goal of establishing a more robust in vivo nerve injury assay to query changes in glial gene expression. In adult flies, axons extending to and from the front, middle, and rear sets of legs project through the prothoracic (PN), mesothoracic (MN), and metathoracic (MtN) neuropil regions of the VNC, respectively. Sensory axons from the

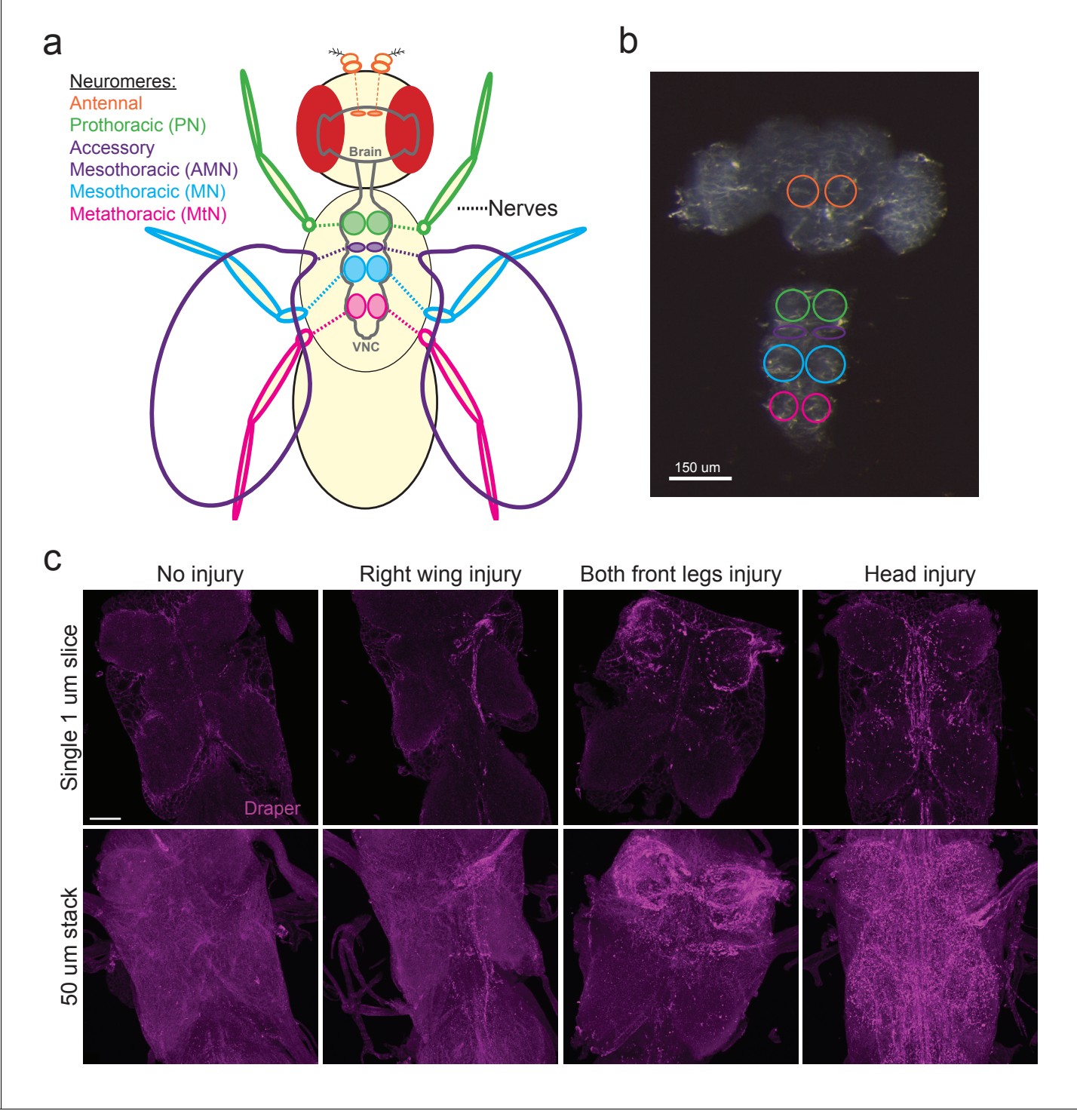

**Figure 1.** The *Drosophila* VNC serves as a new model to study glial responses to acute axotomy. (a) Schematic representation of *Drosophila* brain and VNC showing major neuromere regions as uniquely colored circled regions in the CNS. Color of appendages (antennae, legs, wings) correspond to the neuromere through which the nerves from each structure project. (b) DIC image of adult brain and VNC with the corresponding neuropil regions from (a) highlighted with color-coded circles. (c) Representative Draper immunostainings of uninjured VNC and one day after right wing severing, both front legs severed and head removal. Top images show single one-micron slice, while bottom images display 50 μm maximum intensity projection. Unless otherwise noted, scale bar = 30 μm. Genotypes: *Figure 1c*: w[1118].

wing project through the VNC accessory mesothoracic (AMN) neuropil (*Figure 1a,b*). First, we asked if Draper was upregulated in various axotomy paradigms in the VNC by surgically removing select peripheral tissues and then performing Draper immunostaining on dissected VNCs one day after injury. In uninjured animals, Draper was detectable around neuropil and throughout the cortex of the VNC (*Figure 1c*, uninjured panels). In various injury paradigms, Draper was consistently upregulated selectively on severed nerves 24 hr after injury. For example, removal of a single wing or bilateral ablation of the front legs, resulted in strong Draper immunofluorescence in the neuropil that contained degenerating projections corresponding to each injured structure(s) (*Figure 1c*). Similarly, decapitation elicited robust Draper upregulation throughout the VNC (*Figure 1c*).

The adult *Drosophila* brain contains discrete glial subtypes that vary in location, gene expression, and function (*Awasaki et al., 2008*; *Edwards et al., 2012*; *Freeman, 2015*; *Hartenstein, 2011*; *Omoto et al., 2015*). Cortex glia enwrap neuronal cell bodies throughout the cortex, while neuropil regions contain two unique types of glia: ensheathing glia and astrocytes (*Doherty et al., 2009*). In the adult olfactory system, ensheathing glia, but not astrocytes, respond to degenerating ORN axons by upregulating *draper*, invading the neuropil, and clearing axonal debris (*Doherty et al., 2009*; *Logan and Freeman, 2007*; *Logan et al., 2012*; *MacDonald et al., 2006*). We used several well-characterized glial subtype drivers to express membrane-tethered GFP (*UAS-mCD8::GFP*) and found that the patterns of glial labeling in the adult VNC recapitulated the patterns of the central brain. The pan-glial driver *repo-Gal4* appeared to label most, if not all, VNC glia (*Figure 2a*). The astrocyte driver *alrm-Gal4* also specifically labeled glia immediately surrounding the neuropil and they displayed a classic astrocyte-like, highly branched pattern (*Figure 2a*). Finally, *TIFR-Gal4* labeled glia in an ensheathing-like pattern; we detected high levels of GFP in neuropil-associated ensheathing glia and, as in the central brain, low expression in some cortex glia (*Figure 2a*). Thus, the adult VNC contains similar classes of glia that can be discretely labeled and manipulated with glial subtype-specific genetic drivers.

Draper can be upregulated in either ensheathing glia or astrocytes in response to axon degeneration, depending upon the context (*Doherty et al., 2009*; *Tasdemir-Yilmaz and Freeman, 2014*). To determine if Draper is upregulated in VNC ensheathing glia and/or astrocytes post-injury, we repeated peripheral nerve injuries in flies expressing membrane-tethered GFP in each subtype. Removal of a single wing resulted in an expansion of ensheathing glial membrane and striking upregulation of Draper around the AMN neuropil on the injured side (*Figure 2b*). Notably, these signals appeared to overlap (merged, *Figure 2b*). Ensheathing glial expression of Draper$^{RNAi}$ largely eliminated the ensheathing glial membrane expansion and Draper increase following left wing injury (*Figure 2c*). We did not detect any astrocyte membrane expansion following left wing injury, nor substantial Draper/astrocyte membrane overlap (*Figure 2d*). In addition, expression of Draper$^{RNAi}$ in astrocytes did not attenuate injury-induced Draper upregulation following injury to the left wing (*Figure 2e*). Our results suggest that, as in the olfactory system of the central brain, VNC ensheathing glia, but not astrocytes, respond to degenerating axons in the VNC by upregulating Draper.

## Draper is required for clearance of sensory neuron axons in the adult VNC

Draper is essential for engulfment of degenerating ORN axons in the adult olfactory system (*Doherty et al., 2009*; *Logan and Freeman, 2007*; *Logan et al., 2012*; *MacDonald et al., 2006*). To determine if Draper is similarly required in the VNC, we labeled Gr22c gustatory receptor neurons with membrane-tethered GFP (*Gr22c-Gal4* x *UAS-mCD8::GFP*). A single Gr22c cell body resides on each front leg and projects through the PN neuropil. We transected the right front leg adjacent to the thorax, and imaged Gr22c axons in dissected VNCs after 48 hr. At this time point, no Gr22c GFP+ axonal material was visible in the right (injured) neuropil region, while the contralateral Gr22c projection was intact (*Figure 3a*). Fragmentation of the severed Gr22c axons was blocked by co-expression of Wld$^s$ (*UAS-Wld$^s$*) (*Figure 3a*), indicating that these axons undergo a classic Wallerian degeneration program post-injury. Notably, degenerating Gr22c axons were still detectable in *draper* mutant flies (*Figure 3a*), indicating that Draper is required in multiple regions of the CNS for efficient clearance of axonal debris in adult animals.

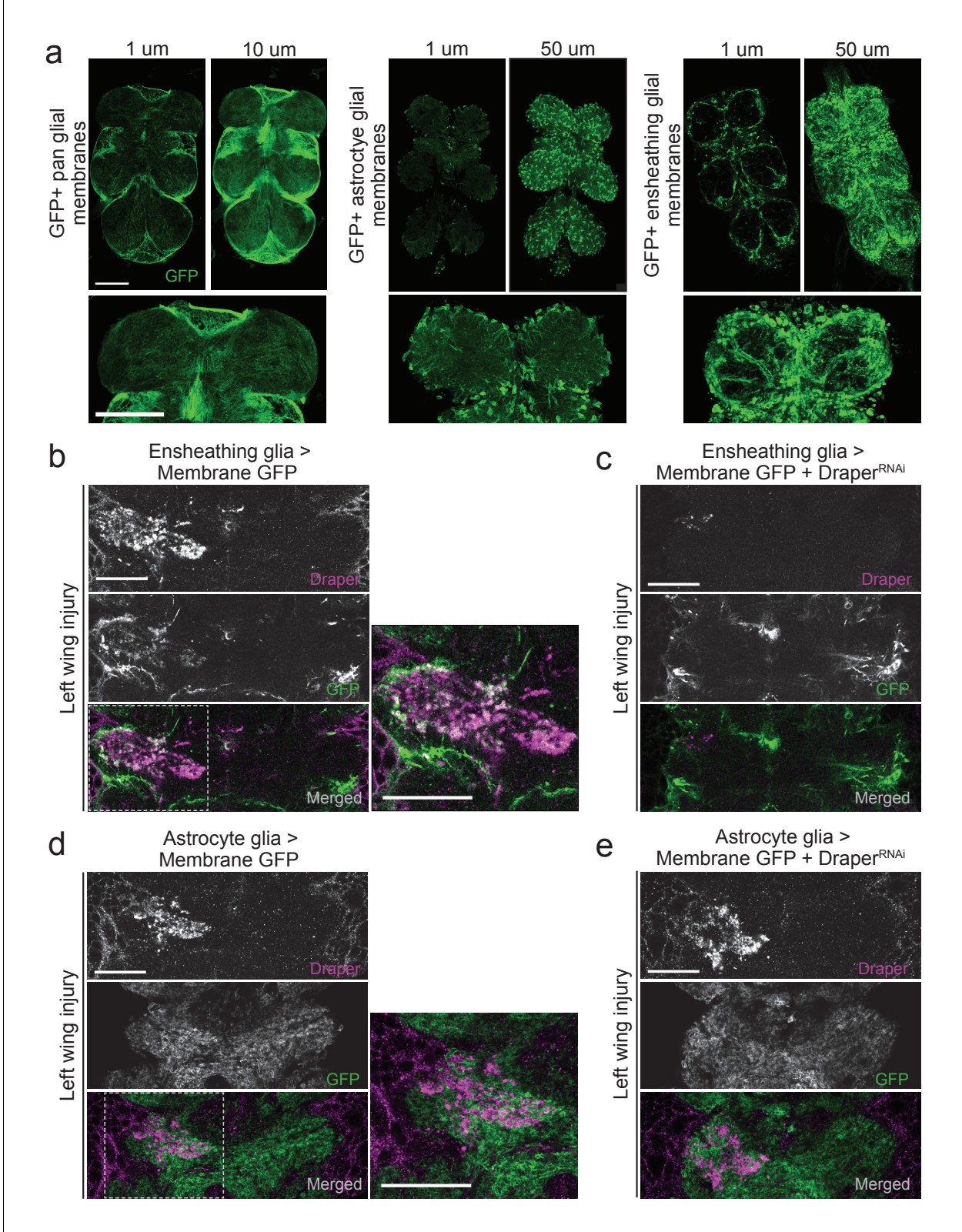

**Figure 2.** Draper and ensheathing glial membranes are recruited to injury sites in the adult VNC after peripheral axotomy. (a) Membrane-tethered GFP (UAS-mCD8::GFP) was expressed under the control of a pan glial driver (left), astrocyte-specific driver (middle), or ensheathing glial driver (right). Images depict single micron slices ~15 μm deep into the VNC. 10 μm (pan-glial) or 50 μm (astrocytes and ensheathing glia) Z-stack projections are also shown. Lower images depict 5 μm (pan glial) or 20 μm (astrocytes, ensheathing glia) higher magnification projections of the prothoracic neuromeres. (b-
*Figure 2 continued on next page*

Figure 2 continued

d) One-micron single confocal slice images showing GFP-labeled glial membranes (mCD8::GFP) and Draper immunostaining of the AMN neuropil region after left wing injury. (b, c) Ensheathing glia expressing mCD8::GFP alone (b) or co-expressing Draper[RNAi] animals (c). (d, e) Astrocyte glia expressing mCD8::GFP alone (b)or co-expressing Draper[RNAi] animals. High magnification merged images in (b) and (d) show regions outlined with dotted line. Scale bars = 30 μm. Genotypes: *Figure 2a*: pan glial: *repo-Gal4,UAS-mCD8::GFP/TM3*. astrocytes: *UAS-mCD8::GFP/CyO; alrm-Gal4/TM3*. ensheathing glia: *UAS-mCD8::GFP/CyO; TIFR-Gal4/TM3*. *Figure 2b*: *UAS::mCD8::GFP/CyO; TIFR-Gal4/TM3*. *Figure 2c*: *UAS-mCD8::GFP/CyO; TIFR-Gal4/Draper[RNAi]*. *Figure 2d*: *UAS-mCD8::GFP/CyO; alrm- Gal4/TM3*. *Figure 2e*: *UAS-mCD8::GFP/CyO; alrm-Gal4/Draper[RNAi]*.

## AP-1 and STAT92e transcriptional activity is stimulated in the VNC after injury

In the olfactory system, ORN axotomy stimulates activity of two highly conserved transcription factors: (1) the Jra/kayak heterodimer, which is homologous to the AP-1 complex (*Macdonald et al., 2013*) and (2) STAT92E (*Doherty et al., 2014*). To first determine if AP-1-dependent transcription is triggered in the VNC after peripheral nerve injury, we used a well characterized Jra/kayak in vivo transgenic reporter (*TRE-eGFP*) to monitor activity of AP-1 in the VNC. *TRE-eGFP* contains 10 tandem AP-1 binding sites upstream of eGFP (*Chatterjee and Bohmann, 2012*). In uninjured VNC, we observed almost no eGFP signal (not shown), but after bilateral ablation of the front legs and wings, we saw robust upregulation of eGFP specifically in the PN and AMN neuropil and surrounding cortical regions (*Figure 3b*). Next, we assessed STAT92E activity in the VNC by using the in vivo reporter *10XSTAT92E-dGFP*, which contains 10 tandem STAT92E binding sites upstream of destabilized GFP (dGFP) (*Bach et al., 2007*). dGFP was undetectable in uninjured VNCs (not shown), but following ablation of front legs and wings, we observed a striking increase in dGFP expression only in the corresponding neuropil regions and surrounding cortex (*Figure 3b*). Together, these results indicate that AP-1 and STAT92E are activated in VNC glia after axotomy and also suggest that common transcriptional programs are stimulated throughout the adult nervous system in response to nerve injury.

## *Draper* is transcriptionally upregulated in the VNC after nerve injury

Doherty et al. recently identified a 2619 base pair (bp) region of the *draper* promoter that is specifically activated in ensheathing glia following ORN injury, referred to as draper enhancer element 7 (*dee7*) (*Doherty et al., 2014*). The *dee7-Gal4* flies express Gal4 under the control of this 2619 bp fragment. Expression is quiescent in the uninjured adult brain, but activated in olfactory ensheathing glia after ORN axotomy, and this promoter fragment contains several requisite STAT92E binding sites for injury-induced activation (*Doherty et al., 2014*). We tested the *dee7-Gal4* driver line in our VNC peripheral nerve injury model by crossing this strain to *UAS-mCD8::GFP* flies. One day after bilateral ablation of front legs and wings, we consistently observed robust activation throughout the anterior VNC (*Figure 3b*), which suggests that similar promoter elements in the *draper* locus are targeted in VNC glia in response to nerve injury. Next, we performed a quantitative PCR (Q-PCR) time course against *draper-I* in VNC tissue 0, 1.5, and 5 hr after ablation of all legs, wings, and/or head. *draper-I* was significantly upregulated in the VNC at all time points in both injury paradigms, with the highest upregulation (~4 fold change) observed 5 hr after all limbs and the head were removed (*Figure 3c*). Collectively, our results indicate that peripheral nerve injury induces transcriptional upregulation of *draper* in the adult VNC and further support the notion that common transcriptional programs are activated in ensheathing glia throughout the adult nervous system.

## Transcriptome profiling of adult VNCs after injury

Once we determined that STAT92E/AP-1 activity, as well as *draper-I* transcription, was increased in the VNC after nerve injury, we assessed genome-wide changes in the adult VNC by performing RNA-seq on dissected VNC tissue after severing legs, wings, and heads of adult animals. Analysis was performed on five biological replicates for each time point (uninjured and five hours post-injury). We obtained 14.5 million ± 1 million single end 100 bp reads per sample (*Figure 4—source data 1*). 85.2 ± 0.39% of the reads mapped to the *Drosophila* genome, with 41.2 ± 2.6% mapping to exons (*Purice et al., 2017*). RNAseq reads were aligned to the *Drosophila* genome (Release 5.57) via the Burrow-Wheeler Alignment tool (*Li and Durbin, 2009*). Raw read counts were converted to transcripts per kilobase million (TPM) (*Wagner et al., 2012*) and genes detected at <2 TPM in both

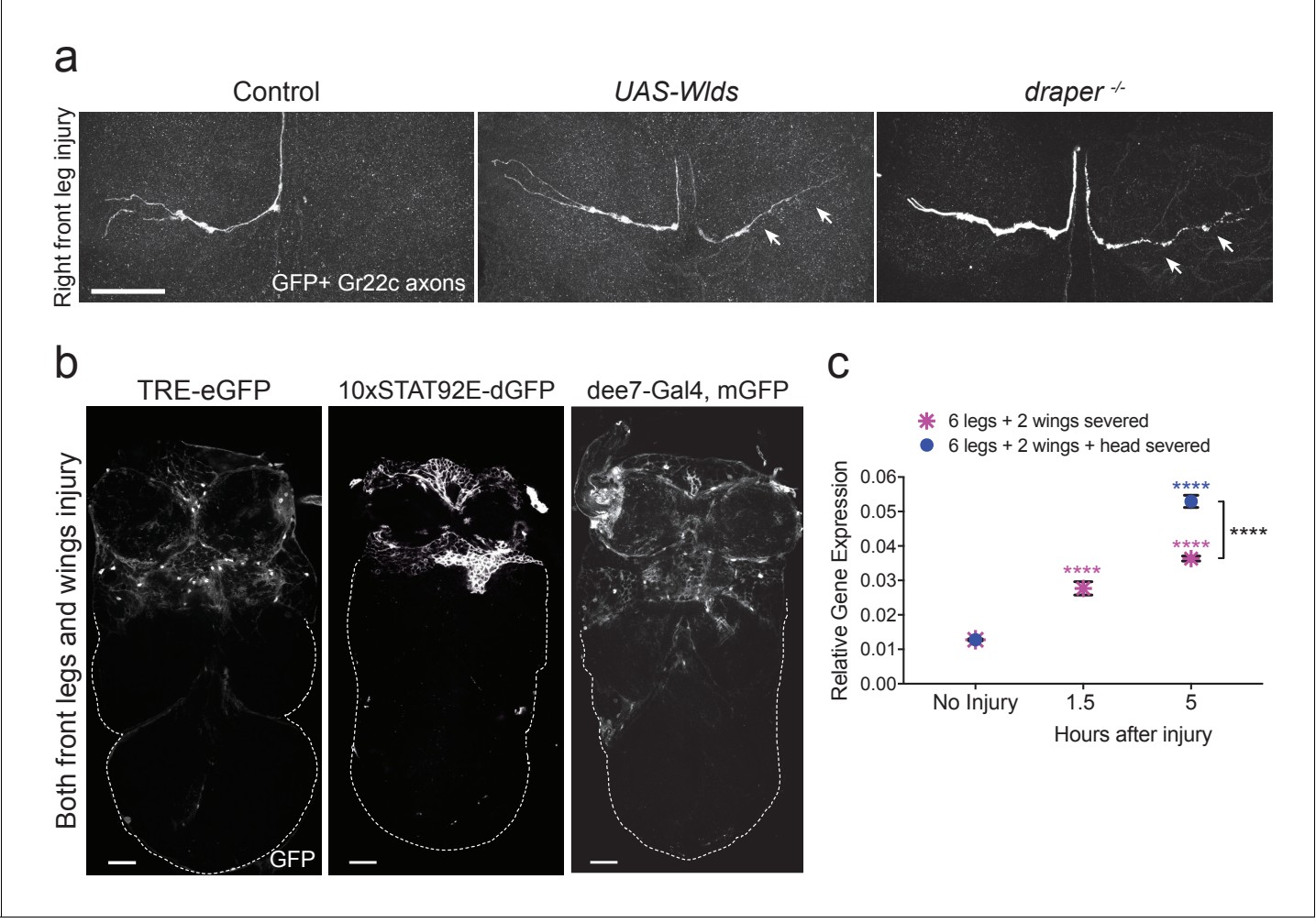

**Figure 3.** Draper is required for clearance of sensory neuron axons undergoing Wallerian degeneration in the adult VNC. (a) Representative confocal images of the prothoracic neuromere (PN) region two days after the front right leg injury are shown. GFP-labeled Gr22c gustatory axons in the PN region of control flies (left), with co-expression of Wlds (center), or in a *draper* mutant background (right). Arrows point to axon that failed to degenerate in Wlds-expressing axons or degenerating axons that remained uncleared in *draper* mutants. (b) Representative one micron images of adult VNCs from flies that carry the AP-1 reporter *TRE-eGFP*, *10XSTAT92E-dGFP*, or the Draper reporter *dee7-GFP*. Reporter activation was largely restricted to the PN and AMN regions after injury to both front legs and wings. Dotted line shows outline of the posterior VNC tissue housing uninjured projections. (c) Quantitative real-time PCR of normalized expression levels of *draper-I* transcript in VNCs following injury to all legs and both wings (magenta asterisks) or following injury to all legs, both wings, and head decapitation (blue circles). Draper threshold cycle (Ct) values were normalized to ribosomal protein L32. Biological replicates: 6 legs + 2 wings: No Injury N = 8; 1.5 hr N = 3; 5 hr N = 3. 6 legs + 2 wings+head: No Injury N = 8; 5 hr N = 7. Mean ± SEM plotted; ****p<0.0001; Two-way ANOVA with Sidak post hoc test. Each injury group was compared to uninjured in the same group. Black asterisks depict comparison between the two injury groups at the 5 hr time point. Scale bars = 30 μm. Genotypes: *Figure 3a*: Control: *Gr22c-Gal4/+; UAS-mCD8::GFP/+*. Wlds: *UAS-Wlds/+; Gr22c-Gal4/+; UAS-mCD8::GFP/+*. *draper-/-*: *Gr22c-Gal4/UAS-mCD8::GFP; draper$^{Δ5rec9}$/draper$^{Δ5rec9}$*. *Figure 3b*: TRE-eGFP: *TRE-eGFP/TRE-eGFP* (on II); 10xSTAT92E-dGFP: *10xSTAT92E-dGFP/10xSTAT92E-dGFP* (on II); dee7-Gal4, mGFP: *dee7-Gal4, UAS-mCD8::GFP/CyO*. *Figure 3c*: *w$^{1118}$*.

conditions (4988 of 14,352) were deemed 'not expressed' (*Wagner et al., 2013*) and eliminated from further analysis (*Figure 4—source data 2*), which was then carried out on 9,364 'expressed' genes. A principal component analysis (PCA) (*Figure 4a*, top panel) and sample distance heatmap (*Figure 4a*, bottom panel) revealed that the variance between biological replicates of the groups (uninjured vs. injured) was greater than the variance within groups. Moreover, a correlation analysis indicated a high degree of correlation between the biological replicates within each sample group (*Figure 4—source data 2*). Analysis of differential expression (DE) was conducted in the START (Shiny Transcriptome Analysis Resource Tool) application (*Nelson et al., 2017*). Using a cutoff of ≥2

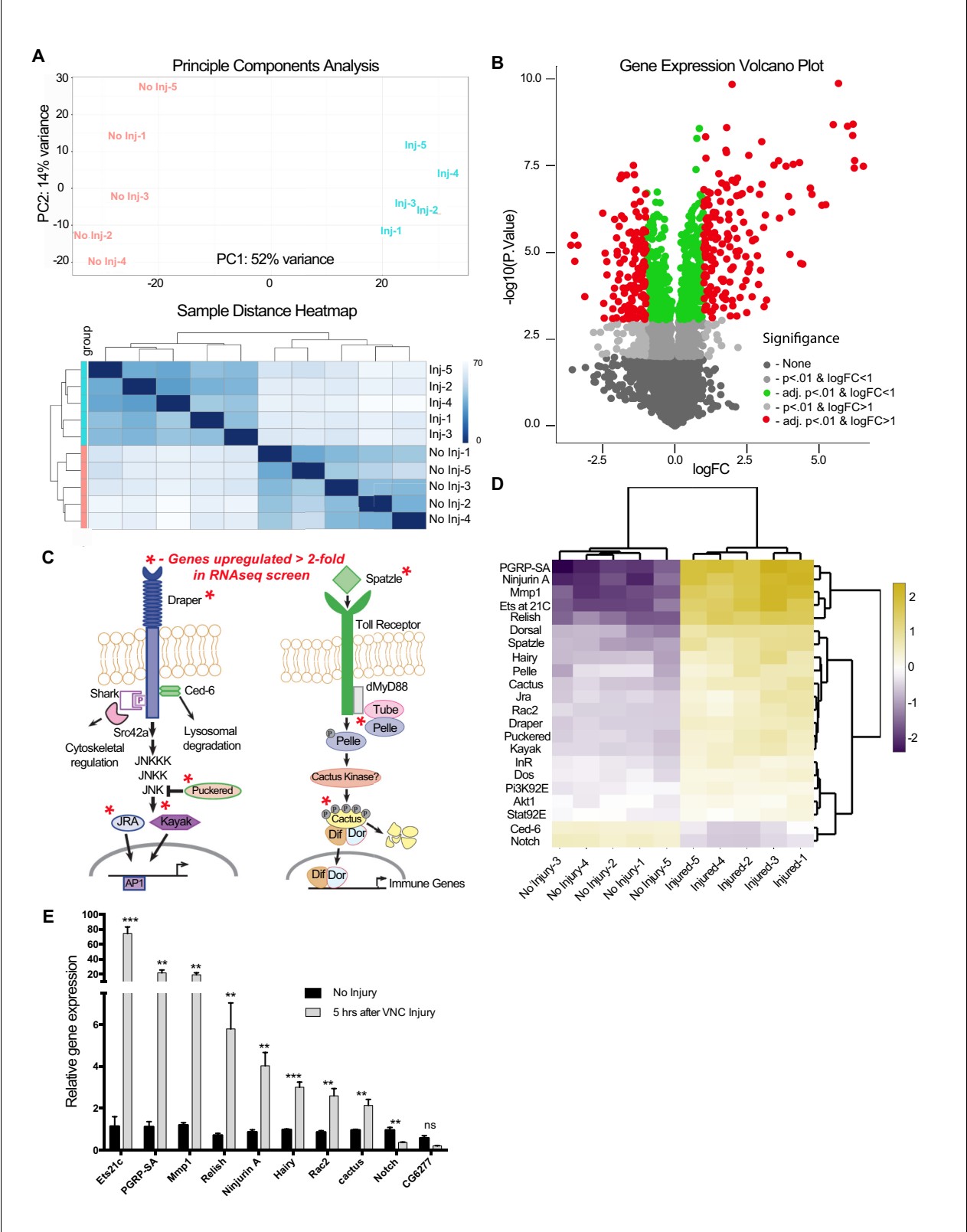

**Figure 4.** RNA-seq analysis reveals functionally discrete groups of injury-induced genes in the adult VNC. (a) A Principal Component Analysis (top) and Sample Distance Heatmap (bottom) of all biological replicates (uninjured and injured) generated in START for genes expressed at >2 TPM (9364 genes). (b) Volcano plot of the gene expression analysis generated in START. The threshold for differential expression was set at 2-fold change (log2 = 1) with an adjusted p-value<0.01. logFC = log of fold change. adj. = adjusted (for multiple comparisons by Benjamin-Hochberg procedure). (c)
*Figure 4 continued on next page*

*Figure 4 continued*

Analysis of injury-induced gene expression revealed known members of the highly conserved Draper/AP-1 and Toll pathways. Genes upregulated ≥2.0 fold in VNC post-injury are noted with red asterisks. (d) START-generated heat map for a small set of hand-selected genes, including genes highlighted in panels (c) and (e), as well as genes previously identified as requisite factors during glial clearance of severed axons (see *Figure 4—source data 5*). Top dendrogram depicts hierarchical clustering of sample distances calculated from gene listed on left side of map. Right dendrogram was generated according to fold-change (uninjured versus injured). Genes at the top of the list were strongly upregulated in injured samples. (e) Quantitative real-time PCR validation of a subset of upregulated or downregulated genes from RNA-seq screen. N ≥ 3 biological replicates; mean ±. SEM plotted; ***p<0.001; **p<0.01; ns = not significant; Students t-test. Genotypes: *Figure 4e*: $w^{1118}$.

The following source data is available for figure 4:

**Source data 1.** RNA assessment, sequencing and alignment metrics.
**Source data 2.** Read counts, Transcripts per Kilobase Million (TPM), biological replicates correlation matrices, and differentially expressed genes.
**Source data 3.** Overall DIOPT analysis for differentially expressed genes.
**Source data 4.** Previously identified *Drosophila* genes functionally implicated in glial responses to axon injury.
**Source data 5.** Comparison of fold change as measured via RNA-seq and RT-PCR methods (Pearson correlation).

fold DE (adj. p-value<0.01, Benjamini-Hochberg procedure), we identified 306 genes (167 upregulated, 139 downregulated) (*Figure 4b* and *Figure 4—source data 2*). Homologs of these differentially expressed genes, which were identified by the DRSC Integrative Ortholog Prediction Tool (DIOPT) program, are provided in *Figure 4—source data 3* (*Hu et al., 2011*). Based on our stringent adjusted p-value of <0.01, we identified an additional 459 genes (225 upregulated and 234 downregulated) that were differentially expressed in the range of 1.2- to 1.9-fold. This gene list is provided in supplemental source data files (*Figure 4—source data files 2–5*), but, because of the low DE values, we have separated this cohort from those genes that fall above the 2.0 DE threshold. We also excluded genes within the 1.2–1.9 DE range from our primary START analysis shown in *Figure 4b* and Annokey analysis (*Table 1*). Because of cell type heterogeneity in VNC tissue samples, genes that were modestly altered (between 1.2–1.9 fold DE) by RNAseq may, in fact, be transcriptionally up/downregulated more robustly (≥2.0 fold) in glia responding to axotomy and represent functionally relevant immunefactors.

## RNA-seq analysis reveals functionally discrete groups of injury-induced genes

Interestingly, components of known innate immunity signaling pathways were included in our list of differentially expressed genes. For example, we detected upregulation of JNK/AP-1 signaling components downstream of Draper, which is consistent with published work describing the role of the JNK/AP-1 pathway during glial recruitment/phagocytic responses to axonal injury in the adult *Drosophila* olfactory system (*Lu et al., 2017*; *Macdonald et al., 2013*) (*Figure 4c,d*). In addition, several members of the Spatzle/Toll pathway were upregulated (*Figure 4c,d*). The Toll pathway, including the downstream transcription factors Cactus and Dorsal, which are known players in innate immunity (*Valanne et al., 2011*), is elicited in response to Wallerian degeneration in vertebrate models (*Boivin et al., 2007*; *Rotshenker, 2011*). Finally, to date, only ~30 *Drosophila* genes have been implicated in glial responses to nerve injury. When mutated, knocked down, or overexpressed in glia, these factors prevent proper clearance of degenerating axons. Nine of these genes (27%) were transcriptionally altered in our RNAseq screen (*Figure 4—source data 4*). Together, these comparisons suggest that our method of transcriptional profiling the adult VNC after peripheral nerve injury is a valuable strategy to identify factors functionally coupled to innate glial immune responses.

Next, we performed confirmational Q-PCR on a subset of differentially expressed genes (8 upregulated and 2 downregulated) from our screen and measured transcript levels 5 hr after severing legs, wings, and heads (*Figure 4e*). With the exception of CG6277, all genes were significantly changed in a manner consistent with our RNA-seq data. Correlation analysis comparing fold-change for all 10 genes (RNA-seq versus Q-PCR) revealed a Pearson's coefficient of r = 0.860 (p=0.003)

**Table 1.** Annokey analysis results of upregulated genes associated with glial membrane expansion and movement. (a) List of terms used in Annokey analysis. (b) Top ten hits associated with our Annokey key terms in *Drosophila*, mouse, and human. MMP-1/MMP-14 (red bold) is included in the top ten list for each species.

| Table 1a | Table 1b | | | | | |
|---|---|---|---|---|---|---|
| **Annokey Search Terms** | **Top 10 Annokey Results** | | | | | |
| [Ff]ilopodia | Drosophila | | Mouse | | Human | |
| [Ff]ilopodium | Gene | Matched Entries | Gene | Matched Entries | Gene | Matched Entries |
| [Ii]nvadopodia | Fasciclin 3 (CADM4) | 131 | Rac1 | 297 | **MMP14** | **709** |
| [Ii]nvadopodium | **Matrix metalloproteinase 1 (MMP14)** | **81** | Tubb3 | 268 | RAC1 | 699 |
| [Pp]odosome(s) | puckered (DUSP10) | 77 | **Mmp14** | **191** | PAK1 | 240 |
| [Mm]igration | lethal (2) giant larvae (LLG1) | 73 | Itga4 | 117 | MYC | 229 |
| ECM | Rac2 (RAC1) | 64 | Nfkb1 | 116 | RELA | 207 |
| [Ee]xtracellular matrix | scab (ITGA4) | 54 | Jun | 89 | JUN | 195 |
| [Ii]nvadosome | rhomboid (RHBDL3) | 52 | Myc | 87 | ITGA4 | 132 |
| [Ii]nvasive | dorsal (RELA) | 51 | Vcl | 83 | TFPI2 | 114 |
| [Mm]etastasize | Jun-related antigen (JUN) | 32 | Ptgs2 | 62 | FLNA | 112 |
| [Mm]etastasis | kayak (FOSL2) | 32 | Flna | 49 | VCL | 53 |
| [Cc]ell invasion | | | | | | |

Source data 1. Human HTML Annokey search results that include hyperlinks to NCBI Gene, GeneRIF and Pubmed databases.
Source code 1. *Drosophila* HTML Annokey results, including hyperlinks to NCBI gene, GeneRIF, and PubMed databases.
Source code 2. Mouse HTML Annokey results, including hyperlinks to NCBI Gene, GeneRIF,pu and PubMed databases.
Source code 3. Human HTML Annokey results, including hyperlinks to NCBI Gene, GeneRIF, and PubMed databases.

(*Figure 4—source data 5*), indicating the robustness of the RNAseq screen to accurately measure the magnitude of differential gene expression.

## RNA-seq reveals a novel candidate transcriptional target implicated in glial migration

Following neural injury, glial cells undergo dramatic changes in size, shape, and migratory behavior to ensure that they quickly infiltrate areas of trauma and efficiently clear degenerating neuronal debris (*Anderson et al., 2014*; *Bardehle et al., 2013*; *Hong and Stevens, 2016*; *MacDonald et al., 2006*; *Napoli and Neumann, 2009*). However, the transcriptional cascades that govern these striking aspects of glial immune responses are not entirely clear. Thus, to glean new insight into the factors required for glial migration and altered membrane dynamics following nerve injury, we took a two-step approach to determine if our VNC injury RNA-seq results might reveal critical conserved genes implicated in cell migration, membrane remodeling, and related events. First, we used the DRSC Integrative Ortholog Prediction Tool (DIOPT) program (*Hu et al., 2011*) to identify the closest orthologs for all differentially expressed genes in mouse and humans (*Figure 4—source data 3*), with the ultimate goal of focusing our analysis on conserved, upregulated genes. Next, we utilized Annokey (*Park et al., 2014*), an open source tool that queries the Entrez Gene database and linked Pubmed articles with a user generated gene list and a set of user defined search terms. For our analysis, we compared the upregulated set of genes (≥2.0 fold, adj p<0.01) and mammalian homologs against a set of ten search terms related to cell motility and morphogenesis (*Table 1a*). The top ten genes with the highest search term matches are listed in *Table 1b*; a complete summary of Annokey matches for all upregulated genes and associated Entrez Gene hyperlinks are provided in *Table 1— Source code 1–3*.

One interesting gene that emerged from this analysis was matrix metalloproteinase-1 (*Mmp-1*, see *Table 1a*), which was a top ten Annokey match to our select search terms across all three species (*Drosophila*, mouse, and human) (*Table 1b*) and was also robustly upregulated in injured VNC tissue by RNA-seq analysis (~12 fold) (*Figure 4—source data 2*). MMPs are proteases implicated in extracellular matrix remodeling, cell migration, and metastasis of cancer cells (*Nagase and Woessner, 1999*; *Rosenberg, 2002*; *Verma and Hansch, 2007*; *Vu and Werb, 2000*). The role of glia-derived MMPs in mammals is controversial. While some studies have shown that MMPs have detrimental roles in response to injury, others show that they are neuroprotective and promote CNS repair (*Zhang et al., 2011*). A detailed analysis of MMP function has been hindered by the fact that there are over 23 partially redundant MMP genes in mammals (*Page-McCaw et al., 2007*). In contrast, the *Drosophila* genome contains only two MMP genes: *Mmp-1*, a secreted protease, and *Mmp-2*, which is GPI anchored (*Page-McCaw et al., 2003*). Therefore, we chose to take advantage of our *Drosophila in vivo* injury models to explore the role of MMP-1 during early responses to nerve axotomy.

## MMP-1 is upregulated in the adult nervous system following axotomy

The expression pattern of MMP-1 in the adult *Drosophila* CNS is unknown. Thus, to first determine MMP-1 protein levels and localization before and after injury in the adult VNC, we used flies that expressed membrane-tethered GFP (*UAS-mCD8::GFP*) specifically in ensheathing glia (*TIFR-Gal4*), performed a unilateral wing ablation, and then stained tissue with α-MMP-1 one day after injury. We observed a robust increase in MMP-1 in the AMN (wing-innervated) neuropil region on the injured side, and a subset of MMP-1 signal clearly overlapped with ensheathing glial membranes (*Figure 5a*, white arrows). MMP-1 was also significantly increased on Western blots of dissected VNC tissue following bilateral ablation of legs and wings (*Figure 5b,c*).

We then focused our analysis on the adult olfactory system, which offers additional genetic tools for cellular analysis. The *Drosophila* olfactory system contains two sets of external structures, the antennae and maxillary palps, which house ORNs that project to the antennal lobes in the central brain (*Figure 5d*). Removal of the antennae or maxillary palps severs the antennal or maxillary nerves, respectively, and local ensheathing glia respond by infiltrating the antennal lobes and clearing degenerating ORN material (*Doherty et al., 2009*). One notable advantage of the olfactory system nerve injury assay is that it permits analysis of degenerating sensory axons projecting into the antennal lobes with no interference from injured efferent projections to the antennae or maxillary palps. First, we performed a small-scale comparative Q-PCR analysis of a few upregulated genes identified in our VNC RNA-seq screen. All three queried genes were significantly upregulated in the VNC after injury (*Hairy*, *MMP-1*, and *Ets21C*) and, importantly, also in the central brain 3 hr after severing the antennal and maxillary nerves (*Figure 5e*), further suggesting that common transcriptional cascades are stimulated in central brain and VNC glia after neural injury.

Next, we assessed MMP-1 levels by immunostaining 24 hr after severing olfactory nerves. In uninjured brains, MMP-1 was detected at low levels on the tracheal network (arrows in *Figure 5f*). MMP-1 levels were dramatically increased in the antennal lobe neuropil regions 24 hr after severing both antennal nerves (*Figure 5f*, middle panel); we also observed a striking increase specifically on maxillary palp glomeruli after bilateral maxillary nerve injury (arrowheads, right panel *Figure 5f*). Together, these findings indicate that MMP-1 is acutely upregulated in the adult olfactory system after axotomy.

Finally, we hypothesized that ensheathing glia are responsible for the MMP-1 increase in the antennal lobes after olfactory nerve injury since (a) we observe a robust overlap between MMP-1 and ensheathing glial membranes in the injured VNC (*Figure 5a*) and (b) based on all known transcriptional and cellular readouts, ensheathing glia are the primary responders to antennal or maxillary nerve injury in the adult *Drosophila* olfactory system. We performed a more detailed analysis of MMP-1 localization in the antennal lobes after olfactory nerve axotomy by using various drivers to express membrane-tethered GFP (*UAS-mCD8::GFP*) in distinct cell types of the antennal lobe, including ensheathing glia (*TIFR-Gal4*), astrocytes (*alrm-Gal4*), a subset of maxillary palp ORNs (*OR85e-Gal4)*, or in all ORNs with the pan-ORN driver (*orco-Gal4*). Notably, after antennal nerve injury, we observed overlap of MMP-1 and expanding ensheathing glial membranes but not astrocytic membranes (*Figure 6a,b*). Moreover, we do not observe MMP-1 accumulation on uninjured

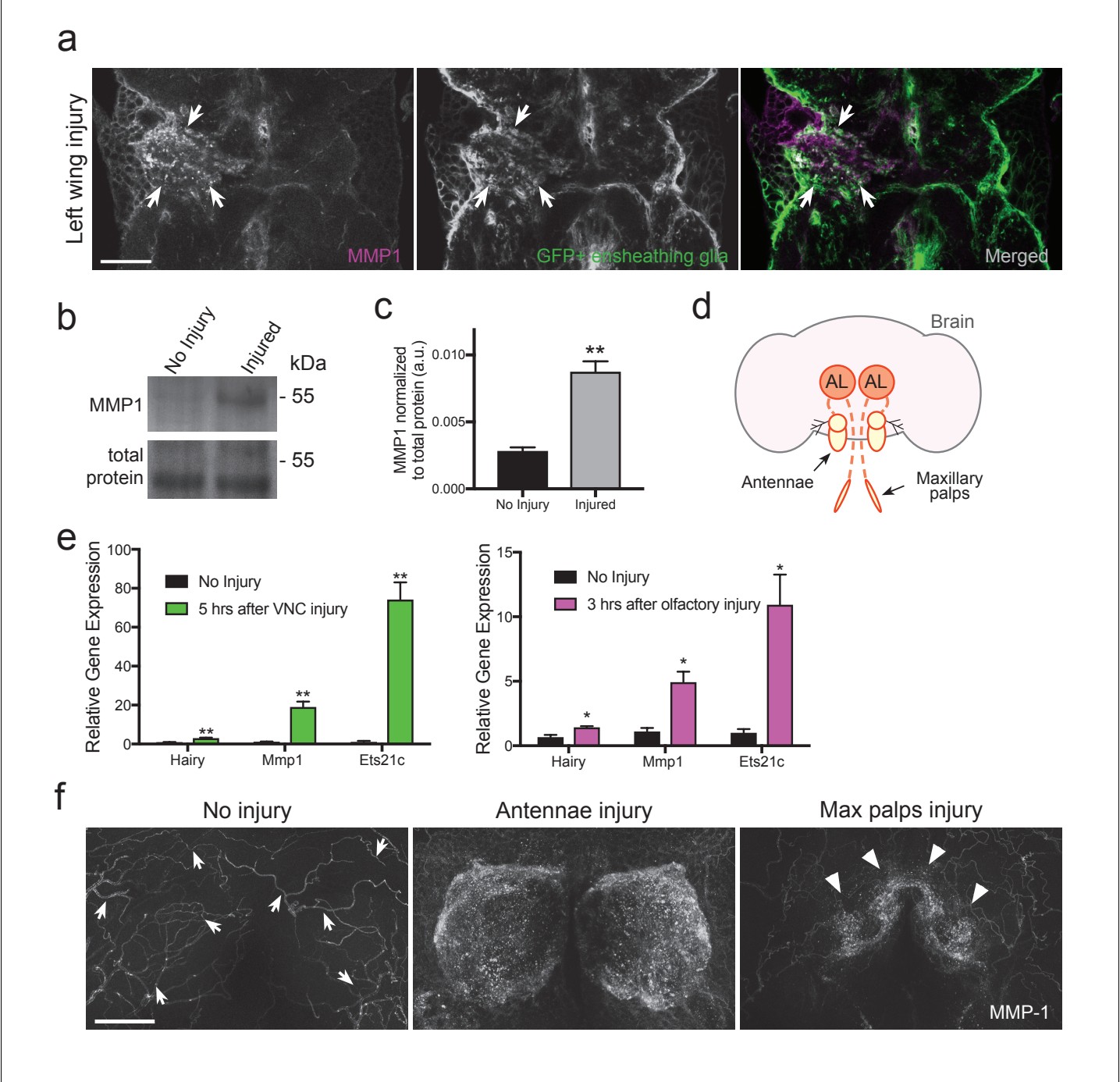

**Figure 5.** MMP-1 is upregulated in the adult VNC and the antennal lobes in response to axon neurodegeneration. (**a**) Representative one-micron confocal images of the anterior VNC expressing membrane-tethered GFP in ensheathing glia and immunostained for MMP-1 after left wing injury. White arrows point regions of increased Draper and glial membrane GFP on the side of injury. (**b**) Western blot stained with anti-MMP-1 (top panel) and total protein stain (bottom). VNC lysates from uninjured and injured (all legs and both wings) $w^{1118}$ flies. (**c**) Quantification of MMP-1 Western blots shown in panel **b**; **$p<0.01$; unpaired t-test; N = 4 biological replicates. (**d**) Schematic representation of the adult olfactory nerve injury assay. Olfactory neurons within the antennae and maxillary palps project into the antennal lobes (dark orange circles labeled as AL). (**e**) Comparative Q-PCR analysis of three select upregulated genes from RNA-seq screen (*hairy*, *MMP-1* and *ets21c*) in injured VNCs (legs, wings, and head) (left) and in the central brain after bilateral antennal and maxillary nerve axotomy (right). Values were normalized to ribosomal protein L32. N = 3 biological replicates per group. Mean ± SEM plotted; **$p<0.01$; *$p<0.05$; unpaired t-test. (**f**) Maximum intensity projections (25 μm) of MMP-1 immunostaining in the antennal lobe region of adult brains. In uninjured animals MMP-1 expression is localized to the trachea (arrows). Robust MMP-1 activity is observed in

*Figure 5 continued on next page*

*Figure 5 continued*

flies after a bilateral antennal injury. Specific accumulation of MMP-1 on maxillary palp axons and glomeruli after bilateral maxillary nerve axotomy indicated by arrowheads. Scale bars = 30 μm. Genotypes: *Figure 5a*: *UAS-mCD8::GFP/CyO; TIFR-Gal4/TM3*. *Figure 5b–f*: *w¹¹¹⁸*.

The following figure supplement is available for figure 5:

**Figure supplement 1.** Uncropped Western blots shown in *Figure 5b*.

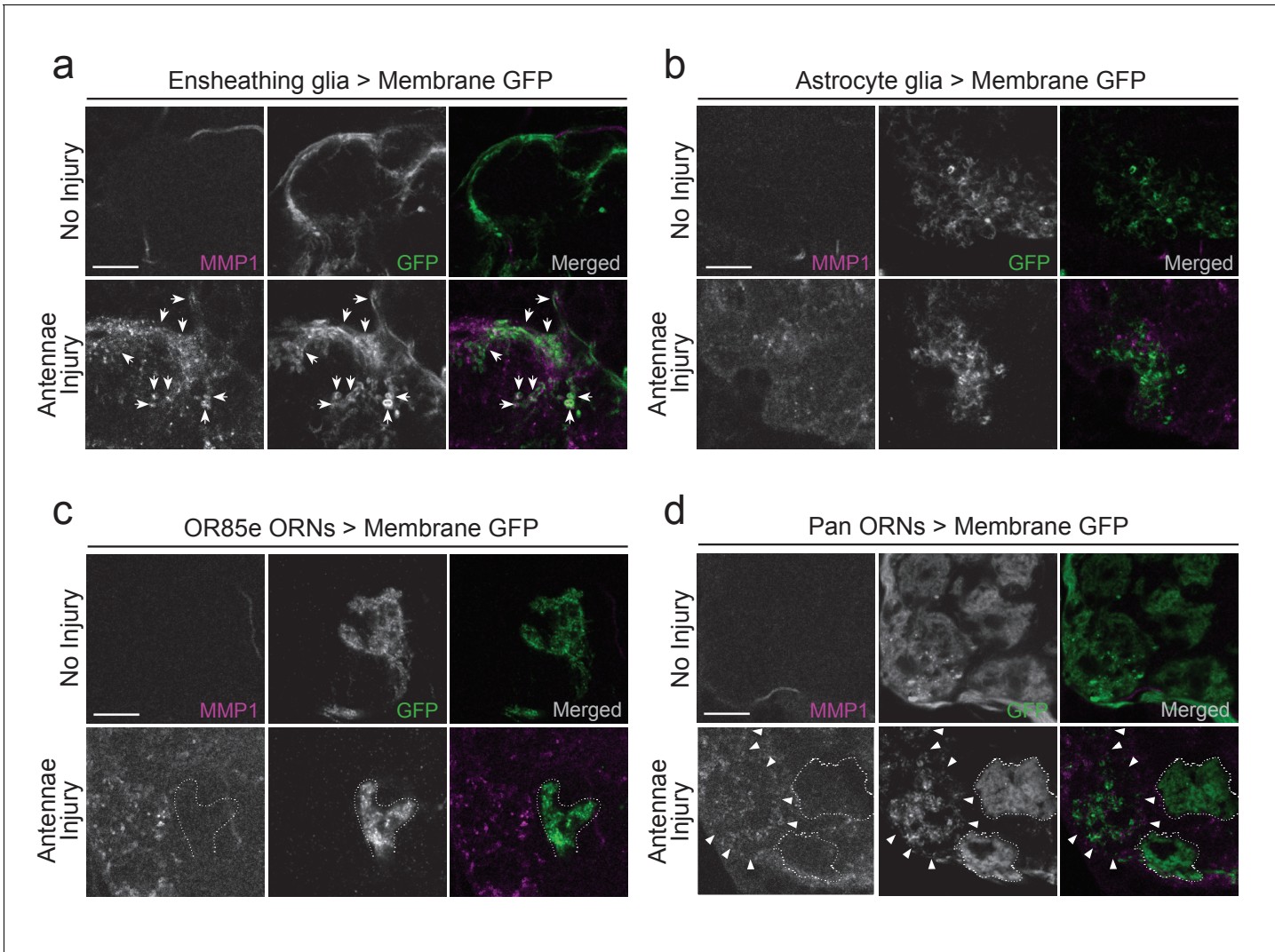

**Figure 6.** MMP-1 largely overlaps with ensheathing glia and degenerating axons after olfactory nerve injury. (a–d) Representative high magnification images of MMP-1 expression three days after antennal nerve injury in flies expressing membrane tethered GFP in discrete antennal lobe cell types. The same antennal lobe region for each uninjured and injured sample is shown. (a) After injury, MMP-1 (magenta) co-localizes with expanding GFP-labeled ensheathing glial membranes (green) (arrows). (b) GFP-labeled astrocyte glial membranes (green) display little overlap with MMP-1 (magenta) after injury. (c,d) After antennal nerve injury, MMP-1 (magenta) is void in glomeruli that contain uninjured GFP-labeled OR85e maxillary palp axons (green, dotted outline). (d) Pan-ORN expression of GFP (green) shows MMP-1 (magenta) accumulation in regions of degenerating ORN material (arrowheads) and lack of MMP-1 in areas that house maxillary palp ORNs (dotted outlines). Scale bars = 10 μm. One micron slice shown for all panels. Genotypes: *Figure 6a*: *UAS-mCD8::GFP/Cyo; TIFR-Gal4/TM3*; *Figure 6b*: *UAS-mCD8::GFP/CyO; alrm-Gal4/TM3*; *Figure 6c*: *OR85e-mCD8::GFP/CyO*; *Figure 6d*: *w¹¹¹⁸*; *orco-Gal4/UAS-mCD8::GFP*.

maxillary palp projections (dotted outlined regions in *Figure 6c,d*) despite the fact they were surrounded by degenerating antennal ORN axons after antennal nerve injury (*Figure 6c,d*).

## MMP-1 upregulation after axon injury requires Draper/STAT92E/AP-1 activity

ORN axotomy initiates a positive autoregulatory feedback loop in ensheathing glia, in which the activated Draper receptor triggers transcriptional upregulation of *draper-I* via JNK/AP-1 and STAT92E (*Doherty et al., 2014*; *Lu et al., 2017*; *Macdonald et al., 2013*). Thus, we wondered if a Draper signaling cascade might also target the *Mmp-1* locus in ensheathing glia after nerve injury. Indeed, we found that the local increase of MMP-1 in the VNC typically observed after leg injury (white arrowheads, *Figure 7a*) was undetectable in *draper* null mutants (*Figure 7a*). Similarly, in the olfactory system, MMP-1 upregulation was completely blocked in *draper$^{-/-}$* flies (*Figure 7b*), indicating that Draper is essential for glial production of MMP-1 in a variety of contexts post-injury.

To determine if the transcription factors STAT92E or AP-1 are required for MMP-1 upregulation after axotomy, we performed in vivo knockdown experiments and assessed MMP-1 levels post-injury. Specifically, we used the pan glial driver *repo-Gal4* to express RNAi against STAT92E (*UAS-STAT92E$^{RNAi}$*) or each subunit of the AP-1 heterodimer, kayak and Jra (*UAS-kay$^{RNAi}$* or *UAS-Jra$^{RNAi}$*). In addition, these flies carried a *tubulin-Gal80$^{ts}$* transgene, which allowed us to temporally control the activity of GAL4 and, thus, specifically express each RNAi construct in post-mitotic adult glia. Interestingly, we found that glial depletion of STAT92E, kayak, or Jra resulted in significant inhibition of MMP-1 induction 24 hr after maxillary nerve axotomy (*Figure 7c–f*). Collectively, these results suggest that axon injury stimulates the Draper receptor to activate *Mmp-1* gene expression in glia, in a STAT92E/AP-1 dependent manner. Aside from the positive transcriptional feedback loop that has been described for *draper* (*Doherty et al., 2014*), *Mmp-1* now represents the first injury-responsive gene downstream of the Draper/MEGF10 receptor, suggesting that a broader Draper/STAT92E/AP-1 transcriptional program is activated in adult glia following axon injury.

## Draper promotes MMP-1 production in ensheathing glia after nerve injury

In the adult *Drosophila* CNS, ensheathing glia, but not astrocytes, express Draper and respond to olfactory nerve axotomy by upregulating *draper-I*, invading injured neuropil areas, and clearing degenerating neuronal debris (*Doherty et al., 2009*). We observed a dramatic increase in MMP-1 staining in a pattern that is strikingly similar to the ensheathing glial membrane expansion (*Figure 2c*). However, as MMP-1 is a secreted molecule, localization of the protein may not reflect its cell of origin. Thus, we took advantage of a previously characterized in vivo transcriptional reporter that expresses cytosolic beta-galactosidase (*β*-gal) under the control of a 4.7 kb fragment of the MMP-1 promoter (*Mmp-1-LacZ*) (*Uhlirova and Bohmann, 2006*). We analyzed activation of this reporter in flies that also carried the STAT92E or AP-1 transcriptional reporters, *10XSTAT92E-dGFP* or *TRE-GFP*, which are both selectively activated in olfactory ensheathing glia after antennal nerve injury (*Doherty et al., 2009*; *Macdonald et al., 2013*). We found that in *Mmp-1-LacZ* flies, *β*-gal was almost undetectable in uninjured brains, but levels were dramatically upregulated around the antennal lobes 24 hr after ORN axotomy (*Figure 8a,b*). Notably, we detected a striking overlap between increased ensheathing glial GFP expression in the *10XSTAT92E-dGFP* and *TRE-GFP* flies and *β*-gal after injury (*Figure 8a,b*). Finally, we used the ensheathing glial driver *TIFR-Gal4* to knockdown Draper by RNAi (*UAS-Draper$^{RNAi}$*) in this glial subset. Strikingly, MMP-1 production was completely blocked one day after antennal nerve injury after depleting ensheathing glia of Draper (*Figure 8c,d*), strongly suggesting that MMP-1 is induced in ensheathing glia following ORN injury, in a Draper-dependent manner.

## MMP-1 is required for dynamic glial responses to axotomy

MMPs can proteolytically target extracellular matrix molecules and influence cell migration, membrane outgrowth and invasion (*Brown and Murray, 2015*; *Nabeshima et al., 2002*). Thus, we reasoned that Draper-dependent activation of MMP-1 may be essential for ensheathing glial cells to access degenerating axons in the antennal lobe neuropil. We expressed membrane-tethered GFP in glia (*repo-Gal4, UAS-mCD8::GFP*) and quantified (a) expansion of glial membranes and (2) MMP-1

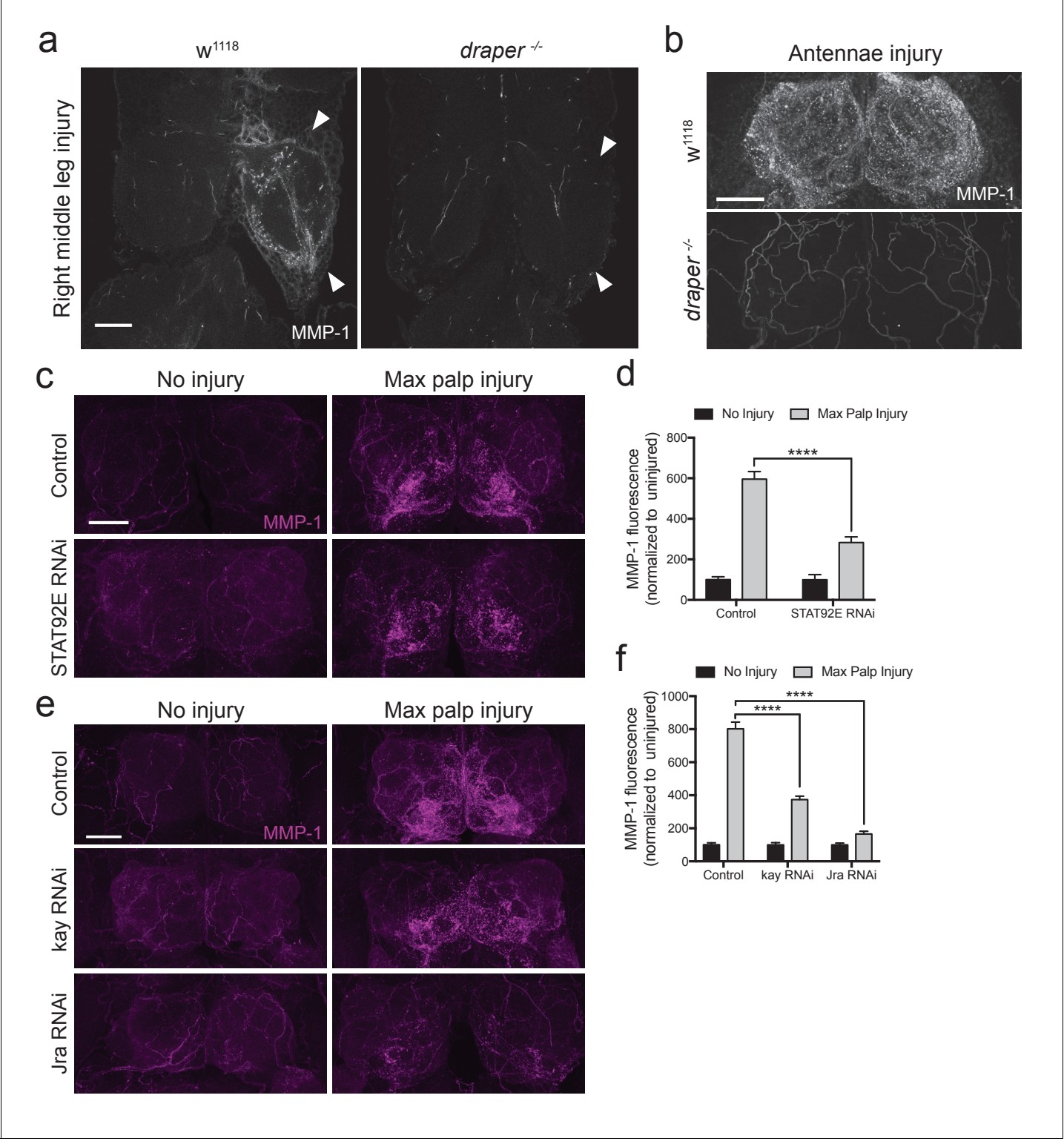

**Figure 7.** Draper/STAT92E/AP-1 are required for MMP-1 upregulation after injury. (**a**) Representative images of control and *draper* null animals one day after right middle leg injury. One-micron slices shown. Arrowheads point to neuropil that contain injured projections. (**b**) Representative images of control and *draper* null animals one day after bilateral antennal nerve axotomy. Maximum intensity projections shown (25 um). (**c**) Adult specific knockdown of STAT92E in glial cells leads to attenuated MMP-1 response to maxillary palp injury. Maximum intensity projections (25 μm) of MMP-1 immunostainings in control and STAT92E[RNAi] flies. (**d**) Quantification of MMP-1 fluorescence in (**c**). Antennal lobes quantified: Control: No Injury N = 17; MP Injury N = 22. STAT92e[RNAi]: No Injury N = 10; MP Injury N = 14. Mean ± SEM plotted; ****p<0.0001; One-way ANOVA with Sidak post hoc test. (**e**)
*Figure 7 continued on next page*

*Figure 7 continued*

Adult specific knock down of kay and Jra in glial cells leads to attenuated MMP-1 response to maxillary palp injury. Maximum intensity projections (25 µm) of MMP-1 immunostainings in control and RNAi-expressing flies. (**f**) Quantification of MMP-1 fluorescence in (**e**). Antennal lobes quantified: Control: No Injury N = 22; MP Injury N = 24. kay$^{RNAi}$: No Injury N = 24; MP Injury N = 28. Jra$^{RNAi}$: No Injury N = 22; MP Injury N = 30. Mean ±SEM plotted; ****p<0.0001; One-way ANOVA with Sidak post hoc test. Scale bars = 30 µm. Genotypes: *Figure 6a,b*: *w$^{1118}$* and *draper -/-: draper$^{Δ5rec9}$/draper$^{Δ5rec9}$*; *Figure 6c–f*: Control: *w$^{1118}$; OR85e-mCD8::GFP, tubulin-Gal80$^{ts}$/+; repo-Gal4/+*. STAT92E$^{RNAi}$: *w$^{1118}$; OR85e-mCD8::GFP, tubulin-Gal80$^{ts}$, UAS-STAT92E$^{RNAi}$; repo-Gal4/+*. kay$^{RNAi}$: *w$^{1118}$; OR85e-mCD8::GFP, tubulin-Gal80$^{ts}$/+; repo-Gal4/UAS-kay$^{RNAi}$*. Jra$^{RNAi}$: *w$^{1118}$; OR85e-mCD8::GFP, tubulin-Gal80$^{ts}$/+; repo-Gal4/UAS-Jra$^{RNAi}$*.

immunostaining around the antennal lobes one day after antennal nerve injury. In injured control flies, we observed a robust MMP-1 increase, particularly at the periphery of the antennal lobes, in areas overlapping with glial cell bodies and notable glial membrane expansion at this timepoint (*Figure 9a,b*, 'AL Border'). Because MMP-1 is a secreted factor, we wondered if the distribution of MMP-1 may change over days. Indeed, three days after antennal nerve injury, we observed more MMP-1 immunostaining localized to the central regions of the antennal lobes ('Inside AL'), as compared to the periphery of the antennal lobes ('AL Border') (*Figure 9a,b*). No MMP-1 expression was detected one or three days after injury in flies expressing MMP-1$^{RNAi}$ under the control of the same glial driver (*Figure 9—figure supplement 1*).

Importantly, we also found that glial membrane expansion was significantly reduced in MMP-1$^{RNAi}$ flies one day after antennal nerve axotomy (*Figure 9a,c*), and several days later we observed striking differences in glial membrane distribution throughout the antennal lobes. In control flies, the glial membrane pattern shifts from distinct wrapping of glomeruli to a more diffuse distribution three days after nerve injury. This shift in pattern is notably less pronounced in glial MMP-1-depleted flies (*Figure 9a,c* and *Figure 9—figure supplement 2*).

Studies of normal cell migration and tumor cell invasion demonstrate that MMP-1 expression influences actin remodeling during cell movement (*Rudrapatna et al., 2014*). We have also previously shown that antennal nerve injury induces striking cytoskeletal remodeling, which requires activity of the protein phosphatase 4 (PP4) complex and likely also the Rho GTPase Rac1 in responding glia (*Winfree et al., 2017*). To determine how glial MMP-1 influences actin polymerization and cytoskeletal remodeling in response to axon injury, we stained control and glial MMP-1$^{RNAi}$ brains with phalloidin-TRITC, which selectively binds filamentous actin (*Wulf et al., 1979*). These flies also *expressed UAS-mCD8::GFP* under the control of *TIFR-Gal4* to visualize ensheathing glial membranes. In control brains, we found that at both one and three days after antennal injury, increased phalloidin levels mirrored that of expanding glial membranes and MMP-1 expression after injury (*Figure 10a,b*). One day after antennal nerve axotomy, phalloidin staining was most robust at the periphery of the antennal lobes, while staining became more notable internally in the neuropil of the antennal lobe after three days (*Figure 10a,b*). Interestingly, ensheathing glial expression of MMP-1$^{RNAi}$ strongly attenuated the increase in phalloidin staining at both time points (*Figure 10a–c*). We propose that reduced phalloidin staining in MMP-1-depleted animals corresponds to attenuated cytoskeletal remodeling within ensheathing glia due to delayed glial infiltration of the antennal lobe neuropil, although we cannot rule out the possibility that secreted MMP-1 also non-autonomously influences actin remodeling in other AL cell types. Nonetheless, our collective results indicate that ensheathing glial generated MMP-1 is essential for proper dynamic glial responses to axotomy in the adult fly brain.

## MMP-1 is essential for proper clearance of severed axons

Because glia must infiltrate the antennal neuropil to access injured axons and efficiently clear degenerating axonal material (*Doherty et al., 2009*; *Logan and Freeman, 2007*; *MacDonald et al., 2006*), we wondered if glial clearance of severed axons would be inhibited by loss of MMP-1. We performed glial specific knock down of MMP-1 in adult flies, using Gal80$^{ts}$ to temporally control activation of *UAS-MMP-1$^{RNAi}$*. These flies also carried an *OR85e-mCD8::GFP* transgene, which labels a subset of maxillary olfactory receptor neurons (OR85e) with membrane-tethered GFP. We performed bilateral maxillary nerve axotomy and quantified GFP$^+$ axonal debris 3 days after injury. Significantly more OR85e axonal material lingered in the brain of MMP-1$^{RNAi}$ flies 3 days after injury

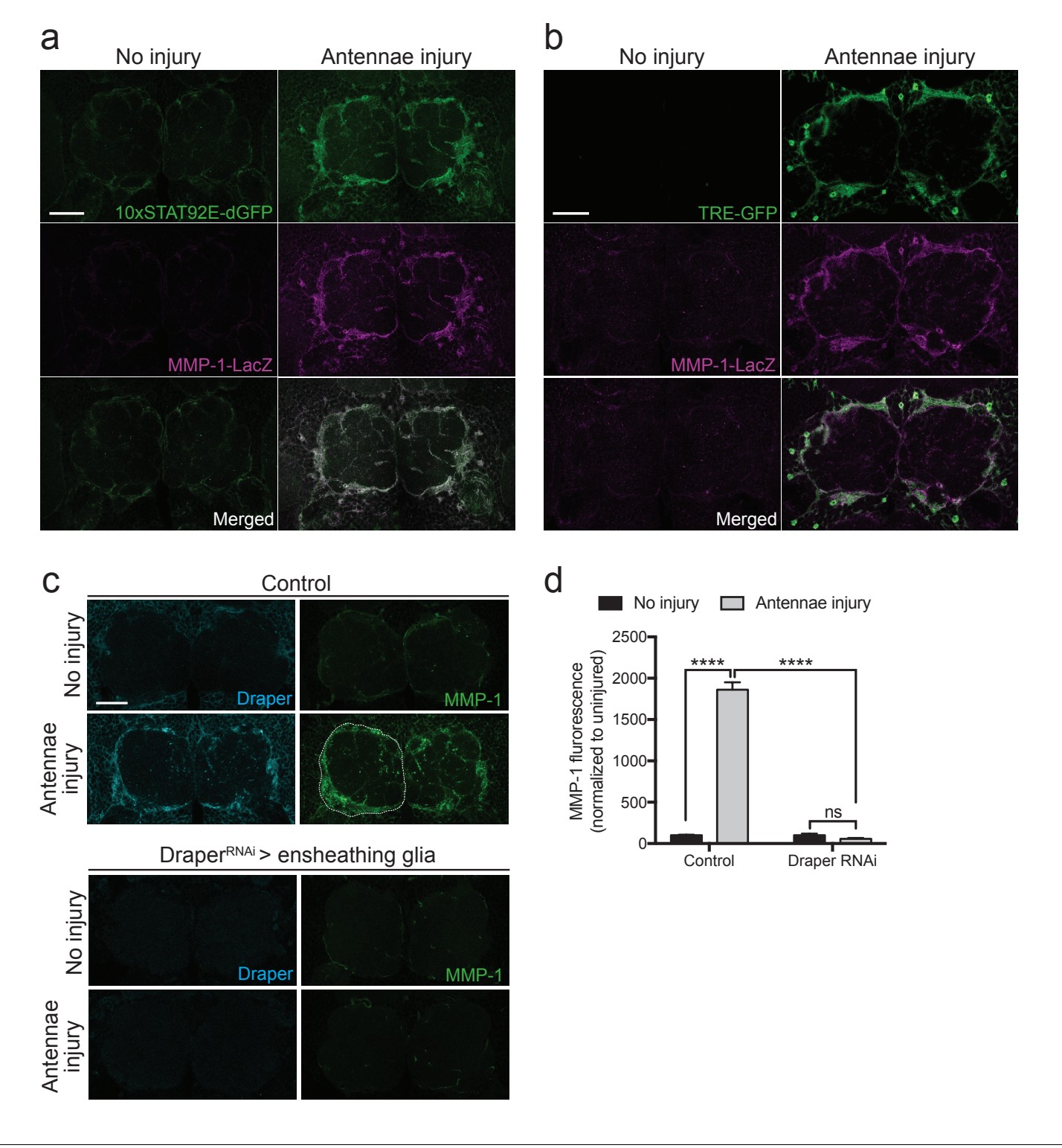

**Figure 8.** Ensheathing glia produce MMP-1 in response to axon injury. (**a**) Representative one micron confocal slices of the antennal lobe region showing STAT92E-dependent activation of dGFP (green) and *MMP-1-LacZ* transcriptional activity (β-gal in magenta) within the same cells one day after antennal nerve injury. (**b**) Representative one micron confocal slices of antennal lobes showing AP-1 activation (green) and *MMP-1-LacZ* transcriptional activity (β-gal in magenta) within the same cells one day after antennal injury. (**c**) Representative Draper (cyan) and MMP-1 (green) immunostainings in control animals or following ensheathing glial knockdown of Draper before and after antennal nerve injury. White dotted outline in control injured MMP-1 panel shows representative ROI for quantification. (**d**) Quantification of MMP-1 before and one day after nerve injury in control and Draper[RNAi]-

*Figure 8 continued on next page*

Figure 8 continued

expressing flies. Antennal lobes quantified: Control: No Injury N = 24; Antennae Injury N = 29. Draper$^{RNAi}$: No Injury N = 12. Antennae Injury N = 24. Mean ± SEM plotted; ****p<0.0001; ns = not significant; Two-way. ANOVA with Sidak post hoc test. Scale bars = 30 μm. Genotypes: *Figure 7a*: *10xSTAT92E-dGFP/MMP-1-LacZ*. *Figure 7b*: *TRE-GFP/MMP-1-LacZ*. *Figure 7c*: Control: *TIFR-Gal4/+*. Draper$^{RNAi}$: *TIFR-Gal4/UAS-Draper$^{RNAi}$*.

(arrowheads *Figure 11a,b*). We also confirmed that MMP-1 accumulation on severed nerves was significantly reduced in MMP-1-depleted animals (arrows *Figure 11a,c*). Finally, to complement our RNAi analysis, we used an independent method to inhibit MMP-1 activity. Tissue inhibitor of metalloproteinase (TIMP) acts as an endogenous inhibitor of MMP-1 by binding to the catalytic domain and inhibiting function (*Brew and Nagase, 2010*; *Page-McCaw et al., 2003*). We expressed *UAS-TIMP* in adult glial cells, quantified OR85e GFP$^{+}$ axons 3 days after nerve injury, and importantly, found that TIMP overexpression also delayed clearance of degenerating axons 3 days after injury (*Figure 11d,e*). Collectively, our results suggest that Draper-dependent activation of MMP-1 is essential for ensheathing glia to invade neuropil regions and properly clear degenerating axonal debris after nerve injury.

## Discussion

The fundamental components of innate glial immune responses to neurodegeneration are strikingly similar in flies and vertebrates. In mammals, acute insults such as spinal cord injury and stroke, activate numerous transcriptional cascades in reactive glia (*Herrmann et al., 2008*; *Kim et al., 2002*; *Park et al., 2016*; *Pennypacker et al., 1994*; *Yu et al., 1995*), but ascribing specific physiological roles to each member of the pathway is an ongoing and challenging endeavor. We have developed a novel in vivo injury model in the adult *Drosophila* VNC and defined a comprehensive dataset of conserved genes that are acutely up- and downregulated in response to nerve injury. This injury-responsive transcriptome now provides a foundation for translatable experiments in *Drosophila* that will rapidly advance our understanding of innate glial immunity mechanisms.

Here, we provide the first in vivo transcriptome profiling analysis in a *Drosophila* nerve injury model. Our assay revealed cohorts of upregulated genes implicated in innate immunity in other phagocytic *Drosophila* cell types and in both professional and amateur vertebrate phagocytes, including members of the Toll-like signaling pathway, which validates the adult VNC injury assay as a reliable model to explore prospective cascades that govern innate glial responses. We should note that because our VNC harvests did not utilize a glial-specific isolation strategy, our dataset likely contains some transcripts upregulated in neurons projecting to peripheral injured structures. Notably, mammalian CNS motor neuron axons do not regenerate after mechanical or ischemic injury unless intrinsic proregenerative transcriptional programs are activated through neuron preconditioning (*Neumann and Woolf, 1999*; *Neumann et al., 2002*; *Qiu et al., 2005*). Similar axonal regeneration programs have been defined in *Drosophila* larval motoneurons, including retrograde pathways that require the highly conserved mediators dual leucine zipper kinase (DLK) and cAMP/PKA (*Brace and DiAntonio, 2017*; *Hao et al., 2016*; *Xiong et al., 2010*). Robust axon regeneration models have not yet been fully developed in adult flies, although some groups have shown that adult *Drosophila* neurons do have some capacity, albeit limited, to regenerate after an injury event (*Ayaz et al., 2008*; *Soares et al., 2014*). It remains to be determined if the adult VNC injury model will reveal transcriptional changes in motor neurons (or other neuronal subtypes) that are components of a preconditioning program.

Our screen revealed MMP-1 as a novel gene upregulated in ensheathing glia in the fly CNS in response to axon degeneration. In addition, we discovered that injury-induced expression of MMP-1 in glia requires the engulfment receptor Draper, as well as the transcription factors AP-1 and STAT92E. Our AP-1 and STAT92E loss of function results mirror findings in other organisms that report MMP genes are regulated by JNK signaling, AP-1 and/or STAT92E (*Chakraborti et al., 2003*; *Korzus et al., 1997*; *Stevens and Page-McCaw, 2012*), highlighting the AP-1/STAT92E/MMP-1 cascade as an evolutionarily conserved module of innate glial immunity. Previous work has demonstrated a requisite role for Draper during infiltration of Drosophila neuropil after olfactory nerve injury (*MacDonald et al., 2006*; *Ziegenfuss et al., 2012*), but the molecular mechanisms were

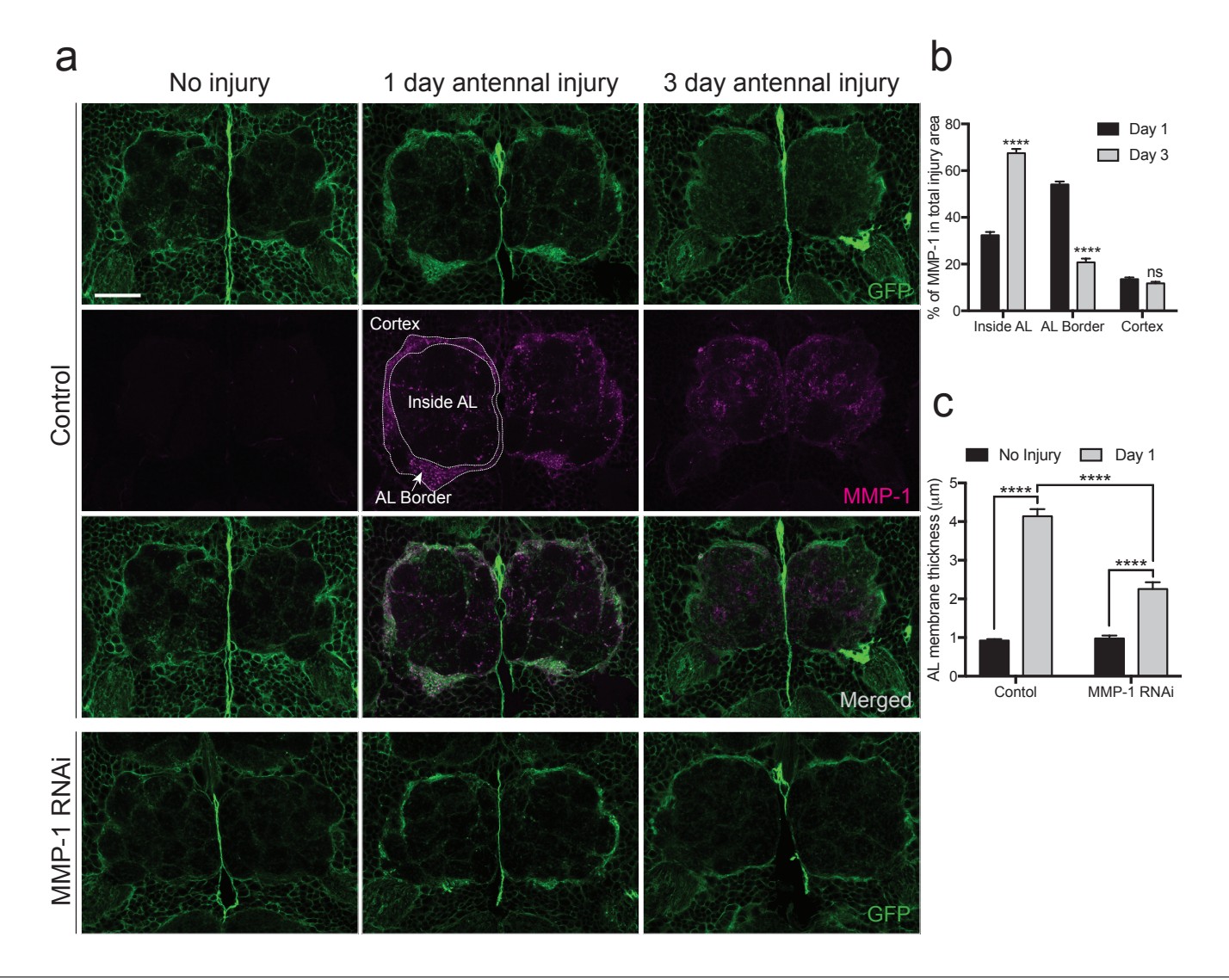

**Figure 9.** Injury-induced glial membrane expansion is attenuated in MMP-1-depleted animals. (a) Glial membranes are labeled with membrane-tethered GFP (green). Representative one micron confocal images of the antennal lobe regions show GFP and MMP-1 immunostaining (magenta) in control and glial-MMP-1$^{RNAi}$ animals. (b) Quantification of MMP-1 in control animals one and three days after antennal nerve injury inside the antennal lobe (Inside AL), at the AL border (AL Border), and in the cortex region (Cortex). Antennal lobes quantified: Day 1: N = 26 antennal lobes; Day 3: N = 21 antennal lobes. Mean ±SD plotted; ****p<0.0001; ns = not significant; unpaired t-test. (c) Quantification of GFP$^+$ glial membrane expansion one day after antennal nerve injury in control and MMP-1$^{RNAi}$ flies; Antennal lobes quantified: Control: No Injury N = 34; Day 1 N = 30. MMP-1$^{RNAi}$: No Injury N = 12; Day 1 N = 24. Mean ±SEM plotted; ****p<0.0001; Two-way ANOVA with Sidak post hoc test. Scale bars = 30 μm. Genotypes: *Figure 8a–c*: Control: *repo-Gal4, UAS-mCD8::GFP/+*. MMP-1 RNAi: *UAS-MMP-1$^{RNAi}$/+; repo-Gal4, UAS-mCD8::GFP/+*.

The following figure supplements are available for figure 9:

**Figure supplement 1.** Adult specific glial expression of MMP-1RNAi leads to efficient glial MMP-1 knockdown.

**Figure supplement 2.** Glial membrane expansion in the neuropil region is attenuated three days after antennal injury in glial knockdown of MMP-1.

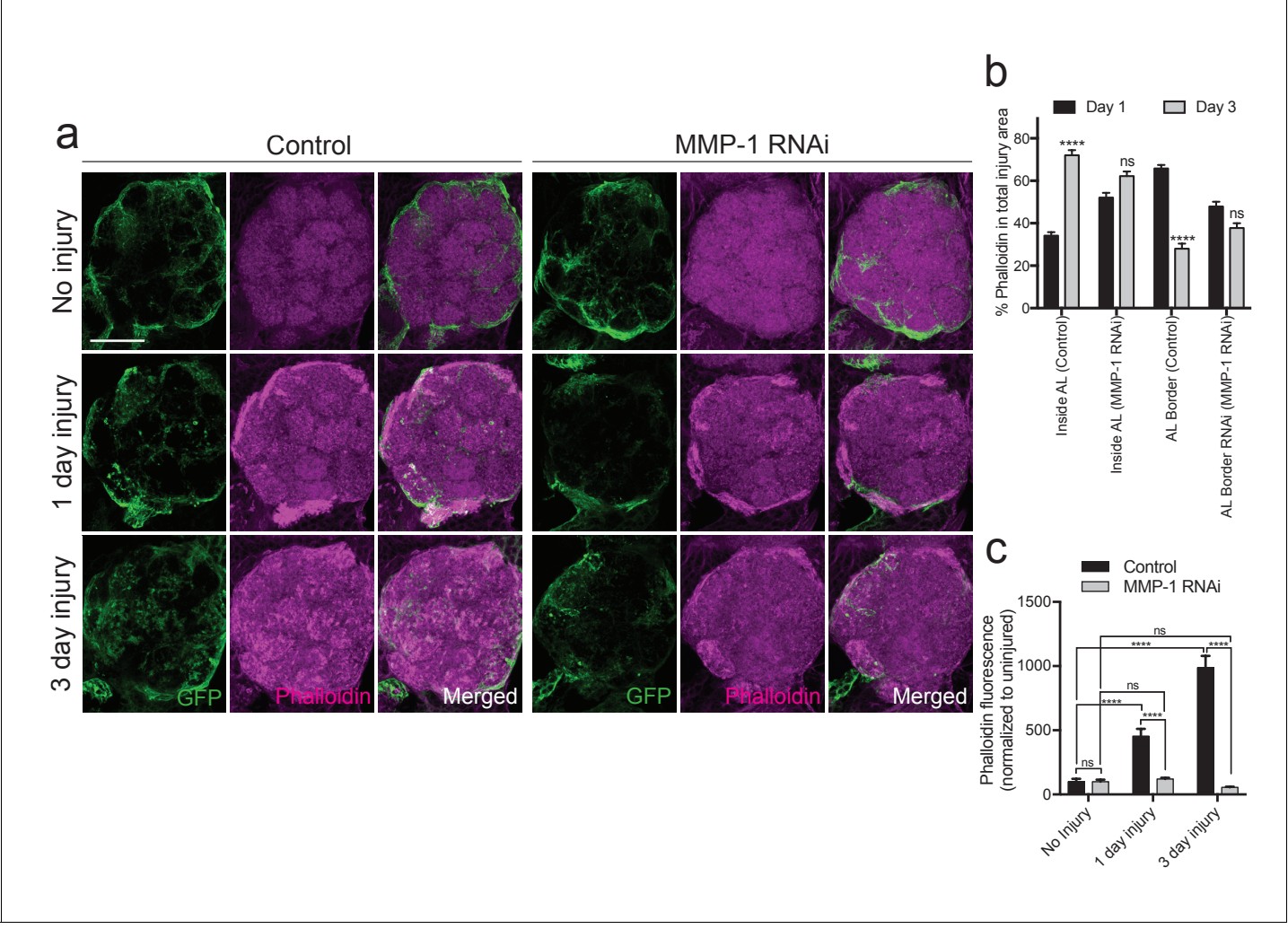

**Figure 10.** Injury-induced ensheathing glial membrane expansion and actin dynamics are attenuated after antennal injury in MMP-1-depleted animals. (a) Representative single antennal lobe immunostainings of Phalloidin-TRITC (magenta) in control and MMP-1[RNAi] animals (uninjured flies and one or three days after antennal injury). Ensheathing glial membranes labeled with membrane tethered-GFP (green). Representative maximum intensity projections shown (5 μm). (b) Quantification of Phalloidin-TRITC expression one and three days after injury inside the antennal lobe and within the AL border in control and MMP-1[RNAi] flies. Antennal lobes quantified: Control: Day 1 N = 18; Day 3 N = 18. MMP-1[RNAi]: Day 1 N = 17; Day 3 N = 12. Mean ± SD plotted; ****p<0.0001; ns = not significant; unpaired t-test. (c) Quantification of total Phalloidin-TRITC within the antennal lobes shown in (a); antennal lobes quantified: Control: No Injury N = 22; Day 1 N = 16; Day 3 N = 18; MMP-1[RNAi]: No Injury N = 18; Day 1 N = 18; Day 3 N = 12. Mean ± SEM plotted; ns = not significant, ****p<0.0001; Two-way ANOVA with Sidak post hoc test. Scale bars = 30 μm. Genotypes: *Figure 9a–c*: Control: *TIFR-Gal4/+*. MMP-1 RNAi: *TIFR-Gal4/UAS-MMP-1[RNAi]*.

unclear. Our work is now the first to couple Draper, which is classically considered an engulfment recognition receptor (*Chung et al., 2015*; *Logan et al., 2012*; *Tasdemir-Yilmaz and Freeman, 2014*; *Zhou et al., 2001*), to a transcriptional cascade that we propose actively promotes extracellular matrix remodeling and glial cell motility via MMP-1 induction. Indeed, MMPs can cleave extracellular matrix molecules to accommodate increased cell migration in a variety of organs and cell types, including metastatic cancer cells (*Al-Alem and Curry, 2015*; *Deryugina and Quigley, 2015*; *Page-McCaw, 2008*; *Thakur and Bedogni, 2016*). MMPs can also target specific ligands or receptors to reveal cryptic sites and modulate intercellular signaling (*Bauvois, 2012*; *Chakraborti et al., 2003*; *Page-McCaw et al., 2007*; *Zhang et al., 2006*). Here, we show that in response to adult antennal nerve axotomy, activation of the ensheathing glial receptor Draper is required for the production of the secreted protease MMP-1 in ensheathing glial cells within 24 hr. Our results also reveal that loss

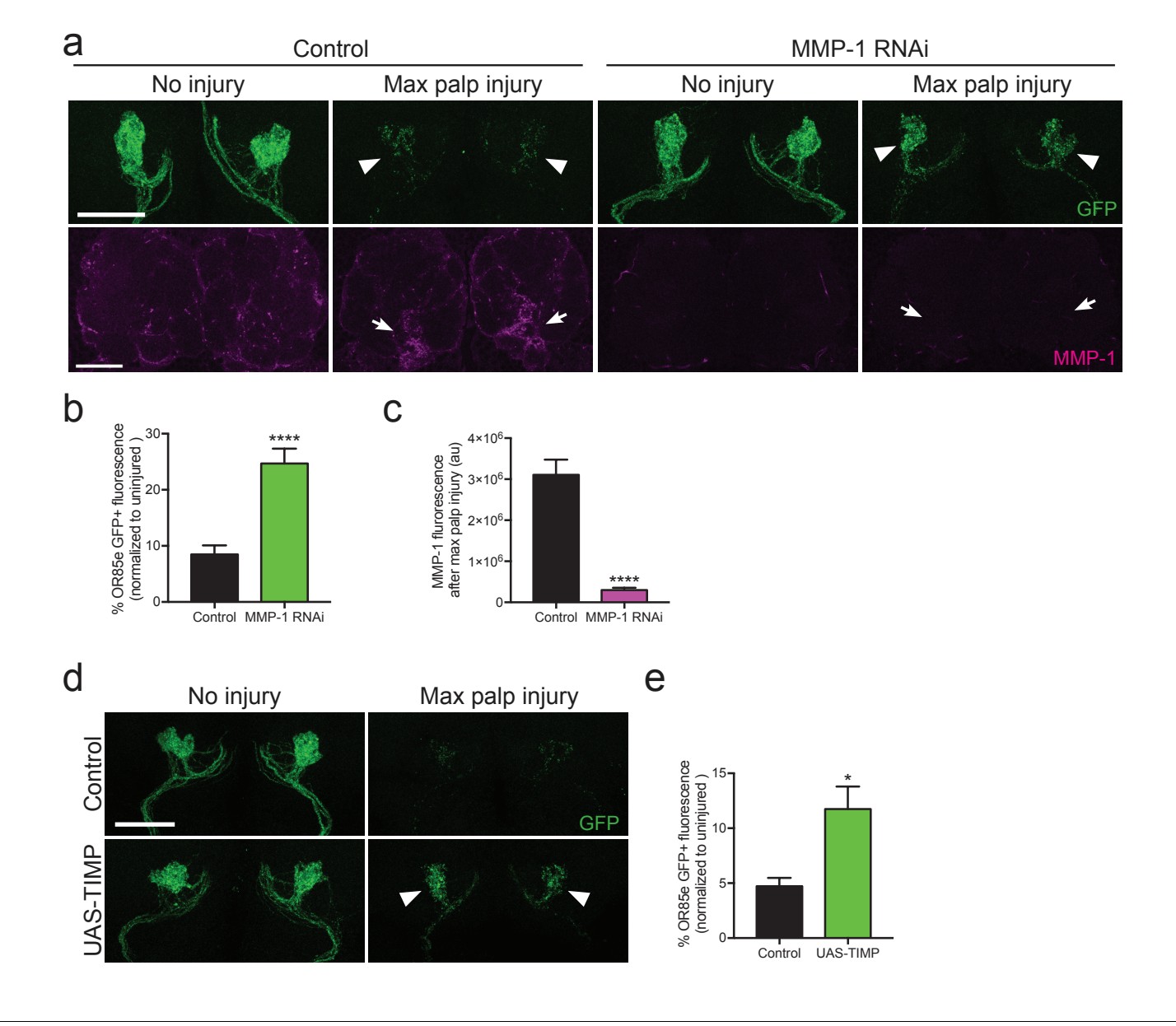

**Figure 11.** Adult specific inhibition of MMP-1 in glia leads to delayed clearance of adult degenerating axons. (**a**) GFP-labeled maxillary ORN axons. (green) before and three days after maxillary palp nerve axotomy in control animals and following adult specific glial knockdown of MMP-1. Arrowheads point to glomeruli quantified in (**b**). MMP-1 immunostainings in same brains (at lower magnification) shown in bottom panels (magenta). Arrows point to OR85e glomeruli regions quantified for MMP-1 fluorescence in (**c**). (**b**) Quantification of axon clearance three days after injury. Antennal lobes quantified: Control: N = 22. MMP-1$^{RNAi}$: N = 30. Mean ±SEM plotted; ****p<0.0001; unpaired t test. (**c**). Quantification of MMP-1 fluorescence in (**a**). Antennal lobes quantified: Control. N = 27; MMP-1$^{RNAi}$: N = 30. Mean ±SEM plotted; ****p<0.0001; unpaired t test. (**d**) GFP-labeled maxillary ORN axons before and three days after maxillary palp nerve axotomy in control animals and adult specific glial overexpression of TIMP. Arrowheads point to perdurance of axon debris in TIMP-expressing flies after axon injury. (**e**) Quantification of OR85e axon clearance three days after axotomy. Antennal lobes quantified: Control: N = 20; UAS-TIMP: N = 32. Mean ±. SEM plotted; *p<0.05; unpaired t test. Scale bars = 30 µm. Genotypes: *Figure 10a*: Control: w$^{1118}$; OR85e-mCD8::GFP, tubulin-Gal80$^{ts}$/+; repo-Gal4/+. MMP-1 RNAi: w$^{1118}$; OR85e-mCD8::GFP, tubulin-Gal80$^{ts}$/UAS-MMP-1$^{RNAi}$; repo-Gal4/+. *Figure 10d*: Control: w$^{1118}$; OR85e-mCD8::GFP, tubulin-Gal80$^{ts}$/+; repo-Gal4/+. UAS-TIMP: w$^{1118}$; OR85e-mCD8::GFP, tubulin-Gal80$^{ts}$/+; repo-Gal4/UAS-TIMP.

of MMP-1 inhibits dynamic membrane expansion of ensheathing glia and delays glial clearance of degenerating olfactory receptor neuron axons. We favor a model in which released MMP-1 remodels the extracellular matrix – and perhaps transmembrane molecules (e.g. adhesion factors) - to facilitate ensheathing glial infiltration into the inner regions of the antennal lobes that house a large number of degenerating axons. This model is supported by the fact that we observe increasing levels of MMP-1 in more central regions of the antennal lobes over time (three days post-injury, as compared to one day post-injury), a time that closely corresponds to ensheathing glia membrane extension and accumulation within the central antennal lobes. However, we can't exclude the possibility that secreted MMP-1 also cleaves factors on degenerating axons to encourage fragmentation or perhaps even release peptide fragments that serve as secondary ligands to further stimulate responses from local ensheathing glia.

MMPs do not appear to cleave target proteins at specific consensus sites. Therefore, identifying the likely abundant MMP targets in the CNS has been an ongoing challenge. One molecule cleaved by *Drosophila* MMP-1 during development is the transmembrane adhesion factor NinjurinA (NinjA). Most organs of the fly contain an intricate tracheal network that transports essential gases or fluids (*Samakovlis et al., 1996*), and MMP-1 cleavage of NinjA on the surface of tracheal cells is essential for proper branching and formation of a mature tracheal system (*Zhang et al., 2006*). Interestingly, Ninjurins are a conserved family of factors that were initially identified by screening for upregulated genes in a rodent nerve injury assay (*Araki and Milbrandt, 1996*). Several groups have since confirmed that one or more Ninjurins are upregulated in injury, stress, and inflammation models, although the specific function of Ninjurins is still unclear (*Jennewein et al., 2015*; *Koike et al., 2008*; *Lee et al., 2016*). Intriguingly, NinjA was also upregulated in our VNC injury screen (*Figure 4d* and *Figure 4—source data 2*), and it will be interesting to explore a potential mechanistic connection between MMP-1 and NinjA in the context of glial recruitment to injury sites and phagocytic clearance of degenerating axonal debris. Finally, the mammalian genomes contain about two dozen MMP genes, and the encoded proteases fall into three different structural categories: secreted, GPI-anchored, and transmembrane-tethered (*Chakraborti et al., 2003*). The significance of spatially restricting select MMPs is still unclear. *Drosophila* MMP-1 is secreted, while MMP-2 is GPI-anchored (*Page-McCaw, 2008*), and it remains unclear whether MMP-2 also contributes to innate glial immunity in the adult fly brain. Nonetheless, the fly clearly offers a powerful model to explore important questions about target specificity and release mechanisms of MMP in reactive glia, as well as in other contexts (*Depetris-Chauvin et al., 2014*).

In addition to influencing disease and cancer progression (*Gialeli et al., 2011*; *Rosenberg, 2002*) and injury responses (*Candelario-Jalil et al., 2009*), MMPs can also influence structural and functional synaptic remodeling in the healthy brain (*Fujioka et al., 2012*). For example, MMP-9 levels are significantly increased in the CA1 region of the hippocampus after induction of late-phase long-term potentiation (L-LTP) (*Huntley, 2012*). Pharmacological or genetic inhibition of MMP-9 blocks not only L-LTP, but also the spine enlargement that is typically associated with persistent LTP. Conversely, gain of function experiments have revealed that MMP-9 is sufficient to promote synaptic structural remodeling, likely through proteolytic cleavage of integrins and/or integrin ligands (*Huntley, 2012*). These recent findings introduce new concerns for how pathological release of MMPs might influence neural circuitry and function. High levels of MMP activity or expression could contribute to maladaptive wiring after acute injury and in disease states. Elucidating the upstream signaling pathways that modulate MMP expression, as well as defining the full array of MMP targets in the healthy and diseased brain, will be critical to parse out the protective versus potentially detrimental role of MMPs in the CNS. Given that the Draper receptor, STAT, AP-1, and MMP family of secreted proteases are all highly conserved, we anticipate that our work will provide new insight into how MEGF10 mechanistically contributes to glial engulfment of apoptotic neurons and synaptic/axonal clearance, as well as functional neuronal recovery, in mammals.

## Materials and methods

### Peripheral nervous system (PNS) axotomy assay

Peripheral nerve injury was induced in adult *Drosophila* by removing legs, wings, and/or head with Vannas scissors (World Precision Instruments # 500260 G, RRID:SCR_008593) while flies were

anesthetized with $CO_2$. Legs were severed at the midpoint of the femur, and wings were severed at longitudinal vein 6. Injured flies were placed dorsal side down on 1% agarose pads in a covered petri dish. In experiments where the head was also removed, the head was removed prior to the legs and wings. We found it was critical to include a wet Kim wipe in the dish to prevent desiccation of the flies. Control flies were also placed in 1% agarose vials and both control and experimental (injured) animals were kept at room temperature for 5 hr. VNCs were dissected in Jan's Saline (0.5 mM $Ca^{2+}$) and immediately frozen on dry ice. Injured animals that did not move the remaining proximal portion of leg in response to gentle forcep manipulation were discarded. For immunohistochemical experiments where only single peripheral organs were removed, the flies were placed back onto food vials after injury.

## Olfactory receptor neuron (ORN) axotomy assay

ORN axotomy was induced in adult *Drosophila* by surgical ablation of the third antennal segments or maxillary palp structures. Flies were maintained at 22–23°C. For adult specific knock down or over expression of genes, flies expressing a temperature sensitive version of Gal80 (*tubulin-Gal80$^{ts}$*) were shifted to 30°C for one week to induce glial expression of each gene of interest and returned to 30°C after ablating maxillary palps until dissection. Control flies for these experiments were treated with the same temperature shift protocol.

## Immunolabeling

Adult *Drosophila* whole flies or heads were fixed (1xPBS, 0.1% Triton X-100, 4% PFA) at room temperature for 16 min. Samples were then washed $1 \times 1$ min and $2 \times 5$ min while rocking in PBSTx0.1 (1xPBS, 0.1% Triton X-100) at room temperature. Fixed samples were maintained on ice while VNCs or brains were dissected at room temperature in PBSTx0.1. Tissue was post fixed for 16 min, washed $2 \times 2$ min in PBSTx0.1, and incubated overnight with primary antibodies in PBSTx0.1. The next day, samples were washed $4 \times 30$ min with PBSTx0.1 and incubated with secondary antibodies (in PBSTx0.1) for 2 hr at room temperature. Samples were then washed $4 \times 30$ min with PBSTx0.1 and mounted on slides in VECTASHIELD mounting media (Vector Labs, RRID:AB_2336789).

## Confocal microscopy and analysis

All samples were imaged on a Zeiss LSM 700 with a Zeiss $40 \times 1.4$ NA oil immersion plan-apochromatic lens. VNCs and brains within a single experiment (i.e. those being directly compared for quantification) were whole mounted under a single #1.5 cover glass in VECTASHIELD. All samples in a given experiment were imaged on the same day with the same confocal microscope settings. Volocity 3D Image Analysis Software (Perkin Elmer, RRID:SCR_002668) was used for fluorescence quantification and GraphPad Prism (RRID:SCR_002798) was used for statistical analysis. Quantification of OR85e GFP-labeled glomeruli was performed on 3D volumes segmented to GFP signal in Volocity. Total intensity measurements were calculated and background fluorescence was subtracted. To quantify MMP-1 levels in adult brains after maxillary palp injury, total intensity measurements were calculated in regions of interest made around the entire antennal lobe. Glial membrane expansion one day after antennal nerve axotomy was quantified by measuring the thickness of GFP$^+$ ensheathing glial membranes at several locations around each antennal lobe on single confocal slices at a consistent anterior depth of 6 μm into the brain.

## Antibodies

Primary antibodies were used at the following dilutions: chicken anti-GFP (ThermoFisher, #A10262, RRID:AB_2534023) at 1:1000; mouse anti-Draper (Developmental Studies Hybridoma Bank, 8A1 RRID:AB_2618106 and 5D14 RRID:AB_2618105) at 1:400, guinea pig anti-Draper (gift from E. Kurant) at 1:10000, mouse anti-MMP-1 (Developmental Studies Hybridoma Bank, 14A3D2 RRID:AB_579782, 3A6B4 RRID:AB_579780, 3B8D12 RRID:AB_579781, 5H7B11 RRID:AB_579779) at 1:50 used at 1:1:1:1 ratio, Phalloidin-TRITC (Sigma, #P1951 RRID:AB_2315148) at 1:250. All secondary antibodies (Jackson Immunoresearch, 715-295-150 RRID:AB_2340831, 703-545-155 RRID:AB_2340375, and 706-605-148 RRID:AB_2340476) were used at a dilution of 1:400.

## Western blot analysis

VNCs were dissected in Schneider's *Drosophila* Medium (ThermoFisher, #21720001) and homogenized in 3 μL 1x Loading Buffer per VNC. Protein lysate of 5–6 VNCs were loaded onto 4–20% Tris-Glycine gels (Lonza, #59517) and transferred to Immobilon-FL (Millipore, #IPFL00010). After transfer, total protein density per lane was measured using MemCode Reversible Protein Stain (Thermo-Fisher, #24585). Blots were probed with mouse anti-MMP-1 (1:100 at a 1:1:1:1 ratio) and incubated overnight at 4°C, washed several times with 1xPBS/0.01% Tween 20, and probed with appropriate fluorophore-conjugated antibodies secondary antibody at 1:2000 (Jackson Immunoresearch, #715-625-150 RRID:AB_2340868) for 2 hr at room temperature. Additional washes were performed with 1xPBS/0.01% Tween 20 and a final wash in 1xPBS. Total protein stain blots were imaged on G:BOX F3 Imaging System and analyzed with ImageJ (RRID:SCR_003070); fluorescent blots were imaged on Li-cor Odyssey CLx (RRID:SCR_014579) quantitative western blot imaging system and data was quantified using LiCor Image Studio software (RRID:SCR_014211). Images in *Figure 5* has been cropped for presentation. Full size image is presented in *Figure 5—figure supplement 1*.

## Sample preparation for RNA-seq

For each biological replicate, 60–80 $w^{1118}$ flies (3–5 days old) with equal numbers of males and females were injured, and 50 VNCs were used per biological replicate for RNA extraction. For injured samples, all six legs, both wings, and the head were severed with Vannas scissors while flies were anesthetized with $CO_2$ as described above. Frozen tissue was crushed in 500 ul of Trizol with glass beads and a pestle and then stored at −80°C until all samples were ready for RNA extraction. A total of 5 biological replicates were collected for each condition (injured and uninjured). RNA extraction was performed by first spinning crushed material at 11,000xg for 10 min at 4°C to pellet cuticle and lipids. Trizol supernatant was transferred to fresh tube, 1/5 vol of chloroform was added to each sample and mixed in a 5' Prime- Heavy Phase Lock Gel, and samples were then centrifuged (12,000xg for 15 min). The aqueous phase was removed and RNA was isolated on RNA Clean and Concentrator−5 columns (Zymo Reserach #R1016). DNAse digestion using Ambion DNA-free kit (ThermoFisher #AM1906) was performed and RNA was quantified using the Qubit fluorometer.

## RNA-seq screen

Total RNA samples were sent to the Massively Parallel Sequencing Shared Resource (MPSSR) at Oregon Health and Science University (OHSU) for library preparation. Briefly, total RNA concentration and sample integrity was assessed using an Agilent RNA 6000 Pico chip on an Agilent Technologies 2100 Bioanalyzer instrument (*Figure 4—source data 1*). Following quantification and quality control, 325 ng of total RNA was subjected to ribosomal RNA reduction via Epicenter's Ribo-Zero Gold kit (Human/Mouse/Rat). The output was then used with Illumina's TruSeq RNA Sample Preparation Kit v2, beginning at the RNA fragmentation step. Poly(A) selection was not performed due to limited starting material. Barcode indexing adapters were ligated and all 10 samples (5 control and five injured) were sequenced (100 bp single end reads) on a single flow cell lane on the Illumina HiSeq 2500 Seqeuncer. Across samples, an average of 85.2% of the fragments mapped to the *Drosophila* genome; 41.2% of these mapped to exons and 9% to introns (*Figure 4—source data 1*). Notably, 22% of reads corresponded to ribosomal RNA (rRNA) and small nuclear RNA (snRNA). Enhanced ribosomal RNA mapping and reduced exon mapping may have been influenced by the fact that our ribosomal RNA depletion kit was not specific to *Drosophila*.

## RNA-seq informatics analysis

RNA-seq alignment was performed with PerkinElmer GeneSifter Analysis Edition (GSAE) software. Briefly, HiSeq 2500 reads were uploaded to Geospiza's server for alignment to the *Drosophila melanogaster* genome (Release 5.57) via the Burrow-Wheeler Alignment tool .*Li and Durbin, 2009* Raw read counts were transformed to transcripts per kilobase million (TPM) (*Wagner et al., 2012*) (*Figure 4—source data 2*) via first dividing by the length of the longest annotated transcript for a given gene and then dividing by the number of mapped reads for that particular biological replicate, thereby normalizing for differences in transcript length and read depth. Before statistical analysis, the gene list was filtered to ensure at least one group (injured or unInjured) had an average TPM value >2. This cutoff has been reported to robustly identify actively transcribed genes

(*Wagner et al., 2013*). This produced a list of 9364 expressed genes that were analyzed for differential expression via the Shiny Transcriptome Analysis Resource Tool (START) application (*Nelson et al., 2017*). Briefly, START performs trimmed mean of M values (TMM) normalization (*Robinson and Oshlack, 2010*), then transforms with mean-variance modelling at the observational level (voom) (*Law et al., 2014*) and then tests for group effects via a linear regression model. Lastly, false discovery rate is corrected via the Benjamini-Hochberg procedure (*Benjamin and Hochberg, 1995*). In our studies, we utilized a conservative adjusted p-value of <0.01 for considering differential gene expression as significant.

Genes that were significantly changed in the injured condition were converted to their closest orthologs in various model organisms using DIOPT (DRSC Integrative Ortholog Prediction Tool) (*Hu et al., 2011*) (*Table 1—source data 1*). Finally, Annokey (*Park et al., 2014*) analysis was performed on upregulated genes to identify factors previously implicated in cell movement, migration, and invasion (*Table 1a, b* and *Table 1—Source code 1–3*).

## Quantitative reverse transcriptase–PCR analysis

Total RNA was extracted and quantified as described for the RNA-seq screen. For cDNA synthesis, 60 ng of DNAse-treated total RNA was reverse transcribed using qScript cDNA SuperMix (Quantabio # 95048–100). The resulting cDNA was diluted 1:5 and 5 ul was used for a single RT-PCR reaction. All real time assays were performed using TaqMan gene expression assays (ThermoFisher) and PerfeCTa FastMix II Rox (Quantabio, #95119–250) on a StepOne Real-Time PCR system (Thermo-Fisher). Ribosomal Protein L28 (Rpl28), TaqMan assay Dm01804541_g1 was used as a control housekeeping gene. Raw Ct values of Rpl28 were unchanged between the uninjured (28.31, SEM. 3037, n = 3) and 5 hr post injury (28.34, SEM. 2030, n = 3) samples. Our RNA-seq results confirmed that Rpl28 is an appropriate housekeeping gene (Digital gene expression levels of Rpl28 (n = 5): No Injury – 77.5 ± 3.14 TPM and Injured – 75.5 TPM ±3.93 TPM.

Additional TaqMan gene expression assays were utilized: Ets21c-Dm01814139_m1; PGRP-SA-Dm01837990_g1; MMP-1-Dm01820359_m1; Relish-Dm02134843_g1; Ninjurin A-Dm01798347_g1; Hairy- Dm01822363_m1; Rac2-Dm01840631_s1; Cactus-Dm01807760_m1; Notch-Dm01841974_g1; CG6277 – Dm02369365_s1.

## *Drosophila* stocks

For all experiments, flies were between 3–10 days old. The following *Drosophila* genetic insertions were used: OR85e-mCD8::GFP/CyO (gift from B. Dickson), UAS-mCD8::GFP (Bloomington 5137, RRID:BDSC_5137), UAS-mCD8::GFP (Bloomington 5130, RRID:BDSC_5130), Repo-Gal4 (*MacDonald et al., 2006*), tubulin-Gal80$^{ts}$ (Bloomington 7108, RRID:BDSC_7108), 10XSTAT92E-dGFP (*Bach et al., 2007*), TRE-GFP (*Chatterjee and Bohmann, 2012*), Gr22c-Gal4 (Bloomington 57605, RRID:BDSC_57605), TIFR-Gal4 (*Yao et al., 2007*), alrm-Gal4 (*Doherty et al., 2009*), Dee7-Gal4 (*Doherty et al., 2014*), Draper$^{Δ5rec9}$ (*MacDonald et al., 2006*), UAS-MMP-1$^{RNAi}$ (*Uhlirova and Bohmann, 2006*), UAS-MMP2$^{RNAi}$ (*Chatterjee and Bohmann, 2012*), UAS-TIMP (Bloomington 58708, RRID:BDSC_58708), w$^{1118}$ (Bloomington 5905, RRID:BDSC_5905), UAS-Draper$^{RNAi}$ (*MacDonald et al., 2006*), MMP-1-LacZ (*Chatterjee and Bohmann, 2012*), UAS-STAT92E$^{RNAi}$ (Vienna Drosophila Resource Center 43866), UAS-kayak$^{RNAi}$ (Bloomington 31391, RRID:BDSC_31391), UAS-Jra$^{RNAi}$ (Bloomington 3159, RRID:BDSC_3159), orco-Gal4 (*Larsson et al., 2004*) (Bloomington 23292, RRID:BDSC_23292).

## Acknowledgements

We would like to thank Dirk Bohmann, Marc Freeman, Bloomington Drosophila Stock Center at Indiana University, the Vienna Drosophila Resource Center, and the Developmental Studies Hybridoma Bank at the University of Iowa for flies and antibodies.

# Additional information

## Funding

| Funder | Grant reference number | Author |
| --- | --- | --- |
| Glenn Foundation for Medical Research | Glenn/AFAR Scholarship for Research in the Biology of Aging | Maria D Purice |
| National Institutes of Health | NIH Neuroscience of Aging Training Grant T32AG023477 | Maria D Purice |
| National Institute of Neurological Disorders and Stroke | R21 NS084112 | Sean D Speese Mary A Logan |
| Fred W. Fields Foundation | New Investigator Award | Sean D Speese Mary A Logan |
| National Institute of Neurological Disorders and Stroke | R01 NS079387-01 | Mary A Logan |
| Medical Research Foundation of Oregon | New Investigator Grant | Mary A Logan |
| Ken and Ginger Harrison Term Professor Award | Faculty Award | Mary A Logan |

The funders had no role in study design, data collection and interpretation, or the decision to submit the work for publication.

## Author contributions

MDP, Conceptualization, Resources, Data curation, Software, Formal analysis, Validation, Investigation, Writing—original draft, Writing—review and editing; AR, Resources, Data curation, Formal analysis; EJM, Data curation, Formal analysis; BJP, DJP, Formal analysis, Investigation, Writing—original draft; SDS, Conceptualization, Resources, Data curation, Software, Formal analysis, Supervision, Funding acquisition, Validation, Investigation, Methodology, Writing—original draft, Writing—review and editing; MAL, Conceptualization, Formal analysis, Supervision, Funding acquisition, Investigation, Methodology, Writing—original draft, Project administration, Writing—review and editing

## Author ORCIDs

Mary A Logan, http://orcid.org/0000-0001-6577-4574

# Additional files

## Major datasets

The following dataset was generated:

| Author(s) | Year | Dataset title | Dataset URL | Database, license, and accessibility information |
| --- | --- | --- | --- | --- |
| Purice MD, Ray A, Munzel EJ, Pope BJ, Park DJ, Speese SD, Logan MA | 2017 | Data from: A novel Drosophila injury model reveals severed axons are cleared through a Draper/MMP-1 signaling cascade | https://www.ncbi.nlm.nih.gov/geo/query/acc.cgi?acc=GSE92759 | Publicly available at the NCBI Gene Expression Omnibus (accession no: GSE92759) |

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
