## [Decision Letter]

Thank you for submitting your article "A novel *Drosophila* injury model reveals severed axons are cleared through a Draper/MMP-1 signaling cascade" for consideration by *eLife*. Your article has been favorably evaluated by Eve Marder (Senior Editor) and three reviewers, one of whom is a member of our Board of Reviewing Editors. The following individual involved in review of your submission has agreed to reveal his identity: Oren Schuldiner (Reviewer #2).

The reviewers have discussed the reviews with one another and the Reviewing Editor has drafted this decision to help you prepare a revised submission.

Summary:

In this paper, Purice et al. examine the molecular mechanisms underlying glial clearance of neuronal debris following neuronal injury. The authors show that a transcriptional program mediated by the cell surface receptor Draper and its downstream target, a STAT protein, occurs following a number of neuronal injury paradigms. They use the prevalence of the response to characterize gene expression changes in nerve cords where most neurons are undergoing degeneration, and identify targets. Analysis reveals that one gene, encoding a matrix metalloprotease MMP-1, is upregulated and secreted by glia. MMP-1 is required for morphological changes in glia, and for efficient engulfment of neuronal debris, and represents the first known functional target of the Draper injury signaling pathway.

How glia react to neuron injury is an important and active field of research as it lies at the heart of development, plasticity, and disease of the nervous system. The work in this paper takes an important step forward in identifying a strategy to explore underlying molecular mechanisms, and in identifying a specific molecular effector. The paper is very clearly written, the experiments appear well controlled, and overall the conclusions justified. However, several points and concerns should be addressed as summarized below.

Essential revisions:

1) The data demonstrate that ensheathing glial MMP1 is required for the clearance of severed axons as well as glial membrane expansion in response to nerve injury. While the data show that it is ensheathing glial MMP that has a regulatory role in the injury response, it appears that by 3 days post injury, MMP1 is not localized to ensheathing glia (Figure 8). It is a bit unclear how the authors are envisioning how MMP1 is acting to regulate membrane expansion. Is the idea that MMP1 is secreted by ensheathing glia and that by 3 days post injury, ensheathing glial MMP1 is binding to/acting on a different cell type? The data seem to suggest that this could be a very likely possibility. To explore this further, the authors should provide some high magnification images at 3 days post injury of MMP1 stains with counter stains for other tissue types like astrocytes, neuronal debris, and healthy neurons.

2) The analysis of glial subtypes is a bit simplistic and could be clarified. Please include a magnified image of the anti-Drpr and CD8 driven by the TIFR and Alrm-Gal4s to provide more convincing evidence that indeed Draper is expressed in ensheathing glia. To strengthen the claim that these cells are ensheathing and not astrocytes, then cell specific RNAi followed by draper staining is needed. Otherwise this should be toned down in the text.

3) Figure 10 is important. However, the statements in the paper claim to have identified a Stat/Drpr/MMP1 cascade – for this, I think it is important to at least try to rescue the *draper^-/-^* phenotype with overexpression of MMP1 by ensheathing glia.

4) A more rigorous analysis of the RNASeq data is needed as well as more details and clarity on methods and decision on cutoffs and better presentation of data is needed. While no additional experiments are necessary, this is important as the RNASeq data are a key part of the paper and would significantly improve the logic of the connection between the two parts of the study.

*Specific comments and suggestions from one reviewer are included below.

Please include the raw data and explain what is a read and other necessary details of method and analyses. For example: how many genes were considered as "expressed" and how was this determined (is one read sufficient? Ten reads?). How were these reads normalized? At low expression levels the noise is very high – therefore, have the authors performed any thresholding to decide which expression data are reliable enough to actually compare – resulting in the total number of genes actually compared (one way to figure out a reasonable threshold would be to scatter plot – on log axes – two repeats and see from what expression levels they actually look similar); all these analyses should result in one big xls file that is well annotated and can be easily read – together, this would be the "data" which would be used for the comparisons.

Comparison of the expression before and after injury: why was 1.2 fold chosen? this seems a bit low and p-values seem small, raising the possibility that the flow of the analysis needs to be reevaluated. All this results in a volcano plot in which the data is spread out and it is thus unclear how many genes are "grey" – thus not statistically changing their expression? This is also evident in [Supplementary-material SD5-data] – all the genes that are expressed in the data set are unregulated. Seems unlikely that there are no genes that are present but not up or down regulated. The Y-axis for the volcano plot needs to be corrected to -log10 (p-value). The KEGG/GO analysis is not informative at all – what do we learn from this?

Table 1 – the Annokey algorithm is not very informative (and therefore not very heavily used.) I understand that the authors want to find a meaningful way to transition from the RNAseq data to MMP1.; however, this transition could be improved. One suggestion is to focus on one group of genes (which should obviously include MMP1, #13 in the entire list) and then check a few candidates and finally focus on MMP1.

[Editors' note: further revisions were requested prior to acceptance, as described below.]

Thank you for resubmitting your work entitled "A novel *Drosophila* injury model reveals severed axons are cleared through a Draper/MMP-1 signaling cascade" for further consideration at *eLife*. Your revised article has been favorably evaluated by Eve Marder (Senior Editor), a Reviewing Editor, and one reviewer.

The manuscript has been improved but there is one remaining issue that needs to be addressed before acceptance that relates to the analyses of the RNAseq data as outlined below.

We suggest considering using a more conventional 2-fold change cut off for the analyses. However if a 1.2 fold change cut off is used, please provide ample explanation and justification as suggested by reviewer #2 below. Given the heterogeneity of the system, it is possible that key genes involved in glial response to injury may only be modestly upregulated in this transcriptional screen, but still be biologically relevant, as pointed out in your response to reviewer comment 2.

*Reviewer #2:*

In this revised manuscript the authors have significantly improved many aspects and have addressed most of my concerns. However, one point still bothers me quite substantially. I even consulted with an expert on analyses of RNA seq and he agreed with me that:

1) If the authors feel very strong about the 1.2 FC then they should explain their rationale within the text. I discourage the authors for going here – as one key reason for the low FC of some genes is the heterogeneity of the sample, in which some people might say – so why didn't they sort the glial cells to get a "tighter" expression pattern.

2) Use a 2FC – in reality, almost all of the genes that the authors want to highlight, like those in Figure 4 ARE induced by a factor of more than two – with the exception of Dor and Ced-6. I think it's a reasonable price to pay. You get a much tighter list of about 350 DE genes. MMP is also unregulated by MORE than 2FC. So I really don't see a good reason why not to opt for this solution.

3) A compromise might be the following: Do most of the analysis as suggested in #2 but then explain that many more potentially DE genes might exhibit a less than 2FC because the tissue is heterogeneous and for that reason, the authors are also providing another list of DE genes with a FC of 1.5 or 1.2 or whatever.

---

## [Author Response]

*Essential revisions:*

*1) The data demonstrate that ensheathing glial MMP1 is required for the clearance of severed axons as well as glial membrane expansion in response to nerve injury. While the data show that it is ensheathing glial MMP that has a regulatory role in the injury response, it appears that by 3 days post injury, MMP1 is not localized to ensheathing glia (Figure 8).*

The reviewer is correct. We do see robust overlap between ensheathing glia (e.g. ensheathing glial driven GFP) and MMP1 at the periphery of the antennal lobes (where the glial cell bodies reside) in the first ~24 hours after antennal nerve injury. This MMP1 immunofluorescence appears to “shift” over days and by 3 days post-axotomy, more MMP1 is detected in interior regions of the antennal lobes. We propose that at day 1, much of the MMP1 signal at the periphery is MMP1 peptide that is being actively translated, transported, and recently released from the glial cells. Over the course of several days, we propose that this secreted factor (MMP1) becomes more apparent in interior regions of the lobes as it precedes thin extending glial membranes in the core region of the lobes. We attempted to demonstrate this shift by comparing MMP1 levels at the border of the antennal lobes (“AL Border”) and in the interior of the antennal lobes (“Inside AL”) (see Figure 9). We acknowledge, however, that this concept could be described in greater depth in the body of the manuscript and have now modified the Results and Discussion to further clarify our observations in this set of experiments and how they are coupled to potential models. We should also note that following glial expression of MMP1^RNAi^, we observe no MMP1 in the antennal lobes 3 days after axotomy (Figure 9—figure supplement 1), which argues against possibility that local antennal lobe neurons (second order projections neurons, interneurons, etc.) are producing MMP1 and serving as the source of MMP1 in the interior regions of the antennal lobes.

*It is a bit unclear how the authors are envisioning how MMP1 is acting to regulate membrane expansion. Is the idea that MMP1 is secreted by ensheathing glia and that by 3 days post injury, ensheathing glial MMP1 is binding to/acting on a different cell type? The data seem to suggest that this could be a very likely possibility. To explore this further, the authors should provide some high magnification images at 3 days post injury of MMP1 stains with counter stains for other tissue types like astrocytes, neuronal debris, and healthy neurons.*

Yes, our favored model is that secreted MMP1 is targeting a key extracellular matrix molecule (or set of molecules) to accommodate additional glial membrane extension into the inner regions of the antennal lobes and more easily access degenerating axons. We cannot exclude the possibility that MMP is instead (or in addition) cleaving factors on the damaged axons to 1) enhance the physical degeneration of the injured nerves or 2) cleave axonal molecules to release additional injury signals that feedback to responding glia. These are all very interesting questions that we hope to investigate in the coming years. We have expanded the manuscript to further describe these models.

We thank the reviewer for the above suggestion. We used various genetic drivers to express membrane-tethered GFP in 1) ensheathing glia, 2) astrocytes, 3) uninjured OR85e maxillary palp axons, or 4) all olfactory receptor neurons. In a new figure (Figure 6), we now show high magnification images of antennal lobe regions showing MMP-1 staining and each GFP-labeled subtype three days after axotomy (and uninjured controls). Briefly, we detect MMP-1 signal overlap with ensheathing glial membranes and actively degenerating axons, but not with astrocytes or intact subsets of olfactory receptor neuron projections – even those that are immediately adjacent to degenerating axons. Please see new version of Figure 6 and corresponding text for further details. We should note that this light microscopy analysis does not provide adequate resolution to definitively state if a small area of MMP1-positive fluorescence is directly associating with the ensheathing glial membrane, axon, or synaptic region. Future efforts that identify and characterize the specific MMP1 cleavage target(s), and where they reside, in this context will give us greater insight into how MMP1 is transported and functioning in the 3D space of the antennal lobe after nerve injury.

*2) The analysis of glial subtypes is a bit simplistic and could be clarified. Please include a magnified image of the anti-Drpr and CD8 driven by the TIFR and Alrm-Gal4s to provide more convincing evidence that indeed Draper is expressed in ensheathing glia. To strengthen the claim that these cells are ensheathing and not astrocytes, then cell specific RNAi followed by draper staining is needed. Otherwise this should be toned down in the text.*

As suggested, we have now provided high magnification images of Draper immunostaining and GFP-labeled membranes (ensheathing and astrocytes) in the injured VNC (new version Figure 2). In addition, we replicated this VNC injury assay (wing injury) in flies expressing Draper^RNAi^ in either ensheathing glia (TIFR-Gal4) or astrocytes (alrm-Gal4) and assessed anti-Draper and GFP signals. Briefly, expression of Draper^RNAi^ in ensheathing glia resulted in almost a complete loss of Draper signal and no notable changes in glial membranes after wing injury. Knockdown of Draper^RNAi^ in astrocytes did not appear to alter Draper upregulation after wing injury. (Please see new Figure 2 and the body of the text for more detail.) These results strongly suggest that ensheathing are responding to peripheral nerve injury, as in the adult olfactory system. Although we cannot rule out an as-yet-unknown response from astrocytes (for which we do not currently possess a readout), we feel our collective results validate a focused analysis on ensheathing glia, with regard to Draper/MMP1 activation and ensheathing glial migration and phagocytosis following nerve injury.

*3) Figure 10 is important. However, the statements in the paper claim to have identified a Stat/Drpr/MMP1 cascade – for this, I think it is important to at least try to rescue the draper-/- phenotype with overexpression of MMP1 by ensheathing glia.*

We have attempted to perform MMP1 rescue experiments using two different strategies. Unfortunately, both approaches have been unsuccessful, and it’s currently impossible for us to provide any conclusive results. First, we tried to co-express UAS-Draper^RNAi^ and UAS-MMP-1 under the control of the glial drivers repo-Gal4 or TIFR-Gal4. These flies also carried the tubulin-Gal80^ts^ transgene so that we could temporally control Gal4 activity and express Draper^RNAi^ and MMP-1 in adult glial cells. Although these experimental flies successfully eclosed as adults, they died with ~24 hours after shifting the temperature of the flies to “turn on” Gal4. This lethality appears to be due to the overexpression of MMP-1 since 1) Draper^RNAi^ can be expressed successfully in glia for weeks in adult animals and 2) control cohorts flies overexpressing glial MMP-1 (lacking the UAS-Draper^RNAi^ transgene) died within a similar timeframe. Unfortunately, a < 24-hour temperature shift in these experimental flies is insufficient to knockdown Draper levels in glia. Next, we attempted to generate new transgenic strains that would ultimately allow us to produce the following set of experimental flies: OR85e-mCD8::GFP,tubulin-Gal80^ts^/UAS-MMP1; TIFR-Gal4, DraperD5rec9/ DraperD5rec9. These flies would allow us to force expression of MMP-1 in adult ensheathing glia in draper null mutant flies and assess clearance of severed olfactory neuron axons. Unfortunately, these experimental flies failed to eclose as adults. It is most likely that this lethality is due to genetic background incompatibility, as opposed to MMP-1 expression.

Although we could attempt this rescue experiment with a fresh approach (new genetic strains, etc.), it would take a substantial amount of time. In addition, we propose that a conclusive result, positive or negative, would not alter our Draper/STAT/MMP1 model. Rescuing axon clearance in draper mutant flies with forced MMP-1 expression would indeed support our findings. We argue, however, that a failure to rescue would not be sufficient to exclude this model. There may be numerous transcriptional targets downstream of Draper that are essential to orchestrate a proper glial immune response. Forced expression of MMP-1 alone may simply not be sufficient to override the loss of the Draper and other downstream factors that are reliant on Draper activation.

*4) A more rigorous analysis of the RNASeq data is needed as well as more details and clarity on methods and decision on cutoffs and better presentation of data is needed. While no additional experiments are necessary, this is important as the RNASeq data are a key part of the paper and would significantly improve the logic of the connection between the two parts of the study.*

**Specific comments and suggestions from one reviewer are included below.*

We thank the reviewer for providing thorough and thoughtful feedback regarding the presentation of our RNAseq results. We have done a substantial re-work of the data and agree that this provides additional key information regarding the analysis and more coherent presentation of our findings.

*Please include the raw data and explain what is a read and other necessary details of method and analyses. For example: how many genes were considered as "expressed" and how was this determined (is one read sufficient? Ten reads?). How were these reads normalized? At low expression levels the noise is very high – therefore, have the authors performed any thresholding to decide which expression data are reliable enough to actually compare – resulting in the total number of genes actually compared (one way to figure out a reasonable threshold would be to scatter plot – on log axes – two repeats and see from what expression levels they actually look similar); all these analyses should result in one big xls file that is well annotated and can be easily read – together, this would be the "data" which would be used for the comparisons.*

To address the above concerns, we have made a number of changes to the analysis and presentation of the RNA-seq screen results:

1) We have now provided data sets with the raw reads and read counts. Specifically, we have supplied an excel file with the raw read counts for all genes from both conditions (uninjured and injured) and all biological replicates. This information can be found in [Supplementary-material SD2-data] and included in our uploaded data to the Gene Expression Omnibus (GEO). The GEO data also includes FASTQ files that contain the raw read sequences. We have provided a private GEO link that the reviewers can use to access this data (https://www.ncbi.nlm.nih.gov/geo/query/acc.cgi?token=oxaraumktfwhlyv&acc=GSE92759). After clicking the above link, enter “GSE92759” into the “GEO accession” box and click the “GO” tab.

2) In order to classify any given gene as “expressed,” we took the following approach: First, to normalize for differences in transcript length and read depth, raw read counts were converted to Transcripts Per Kilobase Million (TPM) (Wagner et al., 2012) by first dividing the raw read counts by the length of the longest annotated transcript for a given gene and then dividing by the number of mapped reads for a given biological replicate (these results are shown in [Supplementary-material SD2-data]). Second, the gene list was filtered to ensure at least one group (Injured or Uninjured) had an average TPM value >2, an appropriate, previously reported cutoff for identifying actively transcribed genes (Wagner et al., 2013). Based on this filtering, 9,364 out of 14,253 genes were deemed “expressed” in our dataset. Notably, we queried the FlyBase RNAseq expression database (http://flybase.org/static_pages/rna-seq/rna-seq_profile_search.html) using a expression threshold level of 1 RPKM (reads per kilobase million), and found that 9,363 and 9,219 genes are expressed in the L3 larval CNS and adult head, respectively, which provides confidence that our calculation of 9,364 expressed genes in the adult ventral nerve cord is a valid value.

3) While exploring the p-values associated with our differential gene expression analysis (addressed below in reviewer comment #2), we opted to use an alternative program, Shiny Transcriptome Analysis Resource Tool (START), to assess differential expression in our uninjured versus injured VNC samples (Nelson et al., 2016). Ultimately, the START analysis identified 348 fewer genes that were significantly changed (as compared to our previous EdgeR analysis). Thus, we have decided to report the more conservative START results in our revised manuscript, however this does not alter any of the key conclusions in the manuscript. In addition to using this new analysis method, we have more clearly explained how the data were analyzed in the Results and Materials and methods sections. We have also added a number of quality control analyses including a PCA analysis, sample distance heatmaps, and a correlation analysis (included in Figure 4 and [Supplementary-material SD2-data]), which more clearly presents the quality of our dataset.

*Comparison of the expression before and after injury: why was 1.2 fold chosen? this seems a bit low and p-values seem small, raising the possibility that the flow of the analysis needs to be reevaluated. All this results in a volcano plot in which the data is spread out and it is thus unclear how many genes are "grey" – thus not statistically changing their expression?*

The choice to use 1.2-fold change as a cut-off is three-fold:

1) Our re-analysis of the data has resulted in a new volcano plot (Figure 4), which addresses the valid concerns raised by the reviewer regarding visibility of the data points and plotting scale. Even with our new analysis (START) and cutoffs for significance, we still detect 169 genes that are significantly changed (green dots in Figure 4: – adj. p-value <.03) but fall below our threshold of 1.2-fold change (log2FC=.263), suggesting that our cutoff of 1.2-fold may be on the conservative side.

2) Published research has identified numerous *Drosophila* genes known to be functionally involved in glial responses to axon injury ([Supplementary-material SD4-data], previously Source Data 5). Our RNAseq analysis revealed that two of these genes were significantly up-regulated in the 1.2-fold range (Stat92e, 1.204-fold; Akt1, 1.253-fold). Moreover, 4 additional genes shown to be essential for proper glial responses to axon injury (InR, Pi3K92E, dos, and ced-6) were upregulated in the range of 1.3 to 1.7-fold. Even the critical glial immune response gene draper, which we know is at least 5-fold upregulated in the ventral nerve cord based on Q-PCR experiments, was only detected as upregulated 2.1-fold in the RNA-seq dataset. Together, these observations suggest that critical genes may only be modestly upregulated in our transcriptional screen.

3) Finally, instead of using an adjusted p-value of 0.05, we have conservatively set the adjusted p-value (Benjamini Hochberg) cutoff to <0.03 to increase our confidence in designating a gene as differentially expressed, despite a smaller fold change in expression.

Regarding our initial p-values reported by the EdgeR analysis and adjusted with Benjamini-Hochberg, we did find that re-analysis with the START program resulted in higher p-values, although the calculated fold change for any given gene are very similar with both pipelines. Overall, despite the shift in p-values, no key results within the manuscript were altered. For example, in comparing our revised [Supplementary-material SD4-data] with the previous version (named Source Data 5 in original submission), the same genes are deemed significantly differentially expressed via both pipelines. The only exception is ced-6, which was accidentally omitted from the original Source Data 5 table in our original submission.

*This is also evident in [Supplementary-material SD5-data] – all the genes that are expressed in the data set are unregulated. Seems unlikely that there are no genes that are present but not up or down regulated.*

We apologize for the confusion prompted by poor labeling in this data file. Now renamed [Supplementary-material SD4-data] in the resubmission, our column label suggested that a subset of genes were not expressed, when our intention was to indicate that they were not differentially expressed in injured VNC samples. This file has now been modified and more clearly labeled.

*The Y-axis for the volcano plot needs to be corrected to -log10 (p-value).*

This label has been changed.

*The KEGG/GO analysis is not informative at all – what do we learn from this?*

In order to conceptually streamline the presentation of the RNA-seq data, we have removed the KEGG/GO analysis.

*Table 1 – the Annokey algorithm is not very informative (and therefore not very heavily used.) I understand that the authors want to find a meaningful way to transition from the RNAseq data to MMP1.; however, this transition could be improved. One suggestion is to focus on one group of genes (which should obviously include MMP1, #13 in the entire list) and then check a few candidates and finally focus on MMP1.*

Indeed, a common challenge with large-scale screens is deciding how to strategically hone in on one factor, or a small set of specific factors, for follow-up in vivo analysis. Some choose to “focus on one group of genes” by choosing, for example, the top 10 most robustly upregulated genes; others may choose to focus on specific signaling pathway of choice. We opted to use Annokey so that we targeted a specific biological process (cell migration/invasion) that was informed by prior biological knowledge and published literature, but not biased by fold-upregulation in our screen, particularly since critical glial factors (e.g. draper) were not robustly upregulated in the RNA-seq dataset. In addition to Mmp-1, we are exploring several additional genes upregulated in our RNA-seq screen and investigating their role during axon degeneration and glial immunity in adult flies, but these preliminary results will form the basis for future manuscripts. We have modified the text in the Results section (which includes Annokey description) in hopes of making the transition from our screen description to our analysis of Mmp-1 in glial responses to nerve injury smoother.

[Editors' note: further revisions were requested prior to acceptance, as described below.]

*The manuscript has been improved but there is one remaining issue that needs to be addressed before acceptance that relates to the analyses of the RNAseq data as outlined below.*

*We suggest considering using a more conventional 2-fold change cut off for the analyses. However if a 1.2 fold change cut off is used, please provide ample explanation and justification as suggested by reviewer #2 below. Given the heterogeneity of the system, it is possible that key genes involved in glial response to injury may only be modestly upregulated in this transcriptional screen, but still be biologically relevant, as pointed out in your response to reviewer comment 2.*

*Reviewer #2:*

In this revised manuscript the authors have significantly improved many aspects and have addressed most of my concerns. However, one point still bothers me quite substantially. I even consulted with an expert on analyses of RNA seq and he agreed with me that:

*1) If the authors feel very strong about the 1.2 FC then they should explain their rationale within the text. I discourage the authors for going here – as one key reason for the low FC of some genes is the heterogeneity of the sample, in which some people might say – so why didn't they sort the glial cells to get a "tighter" expression pattern.*

*2) Use a 2FC – in reality, almost all of the genes that the authors want to highlight, like those in Figure 4 ARE induced by a factor of more than two – with the exception of Dor and Ced-6. I think it's a reasonable price to pay. You get a much tighter list of about 350 DE genes. MMP is also unregulated by MORE than 2FC. So I really don't see a good reason why not to opt for this solution.*

*3) A compromise might be the following: Do most of the analysis as suggested in #2 but then explain that many more potentially DE genes might exhibit a less than 2FC because the tissue is heterogeneous and for that reason, the authors are also providing another list of DE genes with a FC of 1.5 or 1.2 or whatever.*

Reviewer #2’s comments underscore a thoughtful ongoing debate in the scientific community as to how much weight should be applied to various features of comparative transcriptome data, including fold change and *p*-values, to identifying biologically meaningful sets of genes. To address the concerns of reviewer 2, we have made several changes to our data analysis and presentation. First, we repeated our differential gene expression analysis using a more stringent adjusted p value of *p*<0.01, as opposed to our previous threshold of *p*<0.03. We have also parsed out our results such that the results presented in the primary figures of the manuscript utilize a 2.0-fold change cutoff, while differentially expressed genes that fall within the 1.2 – 2.0-fold range are listed separately in supplemental source data files. As noted by the reviewer, because of tissue heterogeneity in nerve cord samples, this cohort of genes (1.2 – 2.0-fold change) may include biologically relevant factors that are

underrepresented (with regard to their fold induction). The fact that our initial positive control (Draper), which is a well-described innate immunity gene, was differentially expressed in our RNAseq screen by 2.09-fold supports this notion. Although we have excluded the 1.2-2.0 FC cohort of genes from the primary analysis, we favor providing this list of genes in an accessible manner to readers who may be interested in exploring their putative role in glial immunity in other systems/contexts.